# PersonaFeedback: A Large-scale Human-annotated Benchmark For Personalization

## Abstract

With the rapid improvement in the general capabilities of Large Language Models (LLMs), LLM personalization, i.e., how to build LLM systems that can generate personalized responses or services that are tailored to distinct user personas, has become an increasingly important research and engineering problem. However, unlike many new challenging benchmarks being released for evaluating the general/reasoning capabilities, the lack of high-quality benchmarks for evaluating LLM personalization greatly hinders progress in this field. To address this, we introduce PersonaFeedback, a new benchmark that directly evaluates LLMs' ability to provide personalized responses given pre-defined user personas and queries. Unlike existing benchmarks that require models to infer implicit user personas from historical interactions, PersonaFeedback decouples persona inference from personalization, focusing on evaluating the model's ability to generate responses tailored to explicit personas. PersonaFeedback consists of 8298 **human-annotated** test cases, which are categorized into easy, medium, and hard tiers based on the contextual complexity of the user personas and the difficulty in distinguishing subtle differences between two personalized responses. We conduct comprehensive evaluations across a wide range of models. The empirical results reveal that even state-of-the-art LLMs that can solve complex real-world reasoning tasks could fall short on the hard tier of PersonaFeedback where even human evaluators may find the distinctions challenging. Furthermore, we conduct an in-depth analysis of failure modes across various types of systems, demonstrating that the current retrieval-augmented framework should not be seen as a *de facto* solution for personalization tasks. All benchmark data, annotation protocols, and the evaluation pipeline will be publicly available to facilitate future research on LLM personalization.

## 1 Introduction

Large Language Models (LLMs) have made tremendous progress in solving a wide range of tasks, from common knowledge understanding to logical reasoning and creative writing (Touvron et al., 2023; Yang et al., 2024b; Liu et al., 2024a; Anthropic, 2023; Jiang et al., 2023; Wang et al., 2024b). These advances have predominantly focused on enhancing the general intelligence of LLMs, often aligning models with universal values and preferences to improve their performance in diverse contexts. However, while these advancements are foundational, they do not inherently guarantee or directly optimize for a crucial aspect of the human-AI interaction: personalization. Personalization refers to the model's ability to tailor its responses to individual users, considering their unique characteristics, preferences, and needs. Despite its potential to enhance user satisfaction and improve the human-AI experience, it remains a nuanced challenge that extends beyond general competency.

This focus on general capabilities is mirrored in the landscape of LLM evaluation. Current benchmarks have largely focused on evaluating models in terms of common tasks such as helpfulness, safety, and alignment with general preferences (Wang et al., 2023; 2024d; Lambert et al., 2024). However, there has been little exploration of how effectively LLMs can generate responses tailored to the pluralistic nature of user personas (Wang et al., 2024c; Salemi et al., 2024b; Jiang et al., 2025). Most existing benchmark (Jiang et al., 2025; Salemi et al., 2024b) rely on deriving implicit user personas from conversation history, assuming that the history of conversations alone is sufficient to understand a user's needs. While this approach is valuable, it is important to note that the ability to infer a user's

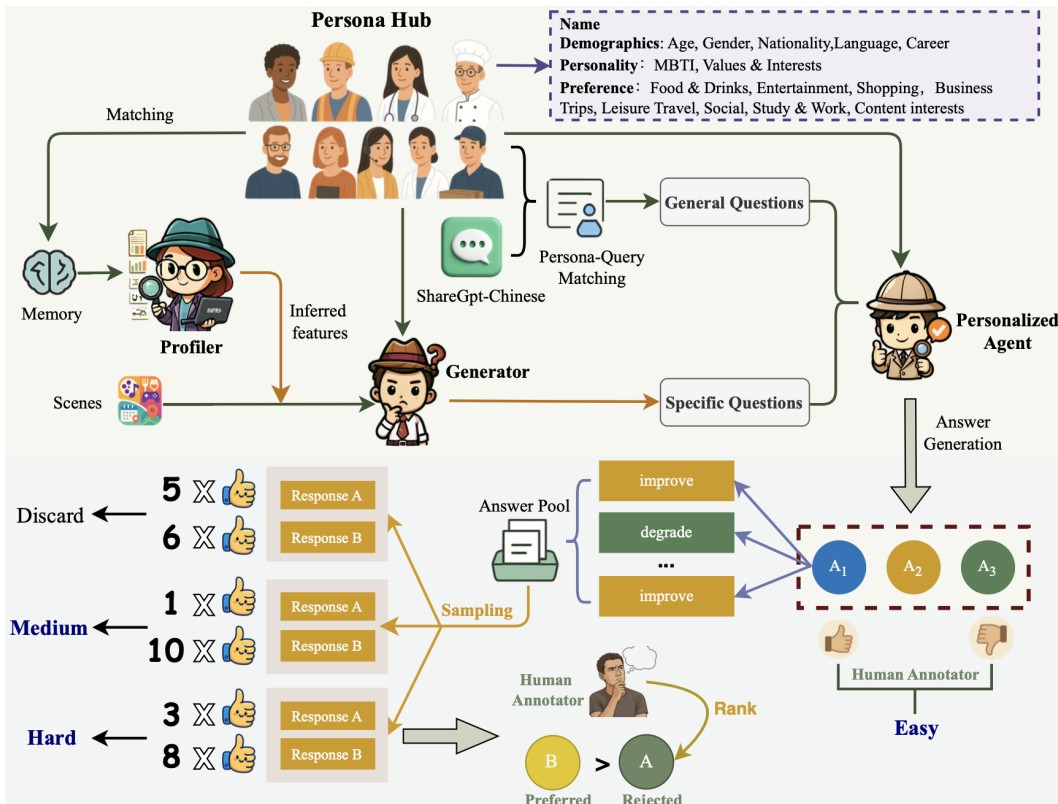

Figure 1: Roadmap of PERSONAFEEDBACK: The Profiler infers user features based on the memory, the Generator generates personalized questions by combining these features with sample scene settings, and the Personalized Agent generates different answers. Finally, PERSONAFEEDBACK is constructed through manual annotation and consistency screening.

persona from chat history and the ability to generate personalized responses are distinct and can be treated as independent tasks.

In this paper, we introduce PERSONAFEEDBACK, a new benchmark that directly evaluates LLMs' ability to provide personalized responses given pre-defined user personas and queries. By explicitly providing the user persona, we separate the task of personalization from persona inference, allowing us to assess how well models adapt their responses to specific user profiles without relying on implicit persona extraction from chat history.

One of the challenges in evaluating personalization is quantifying the degree to which a response is genuinely personalized. Previous framework (Wang et al., 2024c) has attempted to automate this evaluation by adopting the LLM-as-a-judge method to assign scores to responses. Such a method struggles with providing sufficient differentiation and lacks interpretability. To overcome this limitation, PERSONAFEEDBACK introduces a binary-choice evaluation task, where models are presented with two potential responses and asked to choose the one that is more personalized. This approach allows us to effectively measure the degree of personalization by leveraging human-annotated test cases, enhancing evaluation automation, and better quantifying subtle differences in model performance. We illustrate data collecting process in Figure 1.

PERSONAFEEDBACK consists of 8298 **human-annotated** test cases categorized into easy, medium, and hard tiers, based on the contextual complexity of the personas and the difficulty of distinguishing between two personalized responses.

By presenting state-of-the-art (SoTA) LLMs with these test cases, we assess their proficiency in identifying personalized responses in various contexts and across different user profiles. Our empirical results reveal that although these SoTA models demonstrate impressive performance on complex

task-solving benchmarks (Mialon et al., 2023; Cobbe et al., 2021) and RewardBench (Lambert et al., 2024), they still fall short on the hard tier of PERSONAFEEDBACK. This sharp decline in performance highlights the challenges faced by even the most advanced models when dealing with more nuanced personalized scenarios.

By observing the empirical results of our thorough evaluation, we derive several key insights:

1. Enhancing reasoning Does NOT yield to better personalization: Long-reasoning LLMs that excel at solving complex tasks do not show significant advantages over their base models when it comes to personalization.

2. Larger is better: As model size increases, the performance of open-source models improves steadily, with large-scale models showing a clear advantage in handling personalized tasks.

3. Reward models struggle with personalized questions: Although SoTA reward models outperform base models on general questions, they lag behind on user-specific ones.

4. RAG falls short in personalization: As shown in Figure 3, despite being provided with relevant content, RAG setups do not outperform No Persona setups in personalization tasks.

5. Persona learning should be made explicit: As illustrated in Figure 3, explicitly providing user persona significantly boosts performance, suggesting that relying solely on implicit persona inference is insufficient.

## 2 RELATED WORK

**LLMs Personalization.** Personalizing language models (LLMs) has gained attention in areas like recommendation systems, search, and research assistants, aiming to provide tailored responses based on user preferences (Barkan et al., 2020; Woźniak et al., 2024; Dai et al., 2024; Li et al., 2024b; Yang et al., 2023b; Kang et al., 2023; Tan & Jiang, 2023; Tao et al., 2023). Recent efforts have expanded into domains like travel planning (Xie et al., 2024), writing assistants (Mysore et al., 2023), book recommendations (Zhiyuli et al., 2023), shopping advice (Yang et al., 2023a), and programming assistants (Gao et al., 2023).

A common approach is fine-tuning personalized LLMs. For instance, Zhou et al. (2023b) combine persona prediction with response generation, while Tan et al. (2024) apply LoRA (Hu et al., 2021) for user-specific fine-tuning. However, fine-tuning can be inefficient as it requires frequent re-training. In contrast, RAG-based personalized LLMs leverage user-specific data. For example, Salemi et al. (2024b) and Salemi et al. (2024a) introduce pseudo-RAG and retriever optimization, while Li et al. (2023) and Mysore et al. (2023) integrate user-authored documents for prompt augmentation. Yet, input length limitations persist, prompting some studies to summarize user histories for effective personalization (Christakopoulou et al., 2023; Zhiyuli et al., 2023; Richardson et al., 2023).

**Synthetic Persona Generation.** There has been a surge of interest in synthetic persona generation to support personalized LLM research. Ge et al. (2024) proposed persona-driven data synthesis at unprecedented scale via a billion-persona "Persona Hub", enabling large-scale, diverse prompting and downstream tasks. Li et al. (2025) critically examined the risks of bias and methodological gaps in current LLM-generated personas, highlighting limitations in diversity and fidelity. Review works by Argyle et al. (2023) and Castricato et al. further contextualize best practices, challenges, and evaluation methodologies for persona and synthetic data development.

**Benchmarking Personalized LLMs.** Existing LLM benchmarks provide standardized frameworks to assess various capabilities such as coding, task solving, and instruction following (Li et al., 2024a; Mialon et al., 2023; Zhou et al., 2023a). For reward models (RMs), which play a vital role in reinforcement learning from human feedback (RLHF) (Ouyang et al., 2022), benchmarks are also being developed to evaluate their effectiveness in guiding LLM alignment with human preferences (Wang et al., 2024d; Lambert et al., 2024). However, these benchmarks mainly focus on general model capabilities and alignment, rather than personalization.

Recently, a few lines of work have investigated in benchmarking personalization of LLMs. LaMP (Salemi et al., 2024b), utilizes public datasets with user identifiers (Ni et al., 2019; Harper & Konstan, 2015) to mock users' historical interactions and test cases. Zollo et al. (2024) propose to leverage top-ranked reward models and prompt them to simulate different users to evaluate if a

personalized response is preferred by a certain amount of users. AI Persona (Wang et al., 2024c) addresses the lifelong learning process of persona and proposes using LLM-as-a-judge to score the personalization and helpfulness of a response, given a learned user's persona and query. Jiang et al. (2025) proposed PersonaMem benchmark, comprising 7 types of *in-situ* user queries, that focus on benchmarking LLMs' adaptability to ever-changing user persona.

Our PERSONAFEEDBACK explicitly includes user personas alongside queries, enabling a direct evaluation of the model's ability to tailor responses to specific personas, as shown in the table 1.

Table 1: Comparative analysis with existing methods

| Feature | LaMP | PersonaMem | PERSONAFEEDBACK |
|---|---|---|---|
| Explicit Persona Provision | ✗ | ✗ | ✔ |
| Difficulty Levels | ✗ | ✗ | ✔ |
| Human Annotation | ✗ | ✔ (Partially) | ✔ (Fully Annotated) |

## 3 METHODOLOGY

To better evaluate the performance of reward models in personalized interaction scenarios, we propose PERSONAFEEDBACK. The benchmark considers both the degree of personalization and the helpfulness of responses, avoiding excessive personalization that disregards content quality. PERSONAFEEDBACK includes tasks with different difficulty levels (Easy, Medium, Hard) to comprehensively assess the model's personalization capabilities. The final data samples consist of triplets $(P, x, y)$, where $P$ denotes the persona profile, $x$ represents the input prompt for the model, and $y$ is the response generated by the model.

### 3.1 PERSONA CONSTRUCTION

To build realistic and diverse user personas, we first collected 20 real user profiles as initial seeds. These seed profiles contained complete basic elements, including demographic, MBTI, and social background characteristics such as occupations and interpersonal relationships. Based on these real user data, we used seed hints and random combinations of basic elements to further expand the dimensions of user preferences, including daily lifestyle habits (diet, entertainment, shopping), travel patterns (business and leisure), social behaviors, use of productivity tools and content interests. Human annotators review and filter out personas that are overly idealized, inconsistent, or unrealistic. Retained personas have a wide range of occupations, covering major categories such as STEM fields, business, education, healthcare, arts, service industry, and students. The details are referenced in Appendix A and B. Ultimately, we constructed 1,700 personas, of which 200 high-quality personas, manually selected, are specifically used to build the benchmark, while the remaining 1,500 are used for subsequent model training.

### 3.2 QUESTION GENERATION

**Specific Questions** To construct more authentic and diverse user queries while avoiding the issue of generating overly stereotypical and unrealistic questions that often arise from using complete static personas, we adopted the persona learning method from the AI PERSONA (Wang et al., 2024c) framework to dynamically infer user features. Specifically, we define a persona profile as a structured learnable dictionary with fields that include Demographic, Personality, and Preferences.

We have collected open-source data that covers multiple domains, such as social media, reviews, and forums. For each persona, we prompt LLMs to select content that the user is interested in as user's memory data. Then, we randomly sample a set of memory data for each persona, allowing the profiler to infer user features that these contents might reflect. The inferred features are then used by the generator to propose questions.

The generator works as follows: $Q_i = f(P_i, S)$, where $Q_i$ represents the $i$-th generated question, $P_i$ represents the user features inferred from the user's memory data, $M_i$ represents a set of memory data randomly sampled from the persona, and $S$ denotes a list of sub-scenarios randomly sampled following a real user interaction contexts.

Figure 2: An example of an annotated test case from the Specific Easy set of PERSONAFEEDBACK.

Specifically, we annotate 150 questions for 10 real personas along with their corresponding memory data to serve as In-Context Learning (ICL) examples. Then randomly select one question each time to guide the LLM in generating authentic questions. We adopt an embedding model to calculate the similarity score of each question and use a similarity score threshold as a filter to ensure the diversity of our question data. Then we prompt LLM to rephrase the query when characteristic information is directly leaked. Finally, human annotators manually filter out unnatural or unreasonable questions. The details are referenced in Appendix C. After filtering, we obtain over 4,000 user-specific questions.

**General Questions** We sample 30,000 questions from the ShareGPT-Chinese dataset[1] and applied rigorous filtering, retaining only subjective open-ended questions with distinct personal characteristics, while removing objective and factual content answers. After applying length constraints and additional manual screening, we obtain a final set of 1,600 high-quality questions. For each persona, we match queries that closely align with the individual's occupation, background, or personality. Subsequently, we validate these queries to ensure that they are consistent with the persona. This approach finally results in 3841 human-annotated questions.

## 3.3 ANSWER GENERATION

**Answer Generation** We introduce a Personalization Agent and design three distinct answer generation strategies for each question.

- $A_1$: Answers generated using core persona fields (Demographic, Personality) as well as preference traits inferred by the profiler that are directly related to the question.

- $A_2$: Answers generated with 80% of the complete persona profile randomly masked.

- $A_3$: Answers generated based solely on the question itself, without providing any additional persona information.

## 3.4 DATA SELECTION

**Answer Selection** We hire 9 human evaluators to select the response that best matches the profile of a given persona and is the most helpful. The evaluator selects the response that is more consistent with the profile of the persona and more helpful based on the evaluation criteria (see Appendix D). The answer selected by the majority of evaluators serves as the ground truth. Figure 2 presents an example of a selected test case by human labeler.

**Difficulty Levels** Based on the ground truth selected by human evaluators, we construct evaluation tasks at three difficulty levels. For each question, multiple answers are generated using four different models to provide varying degrees of personalization and helpfulness, forming an answer pool. Answer pairs are then randomly sampled from this pool, and nine human annotators are asked to select the better answer for each pair. The difficulty levels are assigned according to the observed consistency among annotators (e.g., proportion of majority votes):

---

[1]Original dataset please refer to https://huggingface.co/datasets/FreedomIntelligence/sharegpt-chinese

- **Easy**: Each task compares the human-selected ground truth with an answer that is generic or does not match persona characteristics, focusing on obvious personalization signals.

- **Medium**: Answer pairs where most annotators' choices are highly consistent are classified as medium difficulty, indicating relatively clear differences between answers.

- **Hard**: Answer pairs with lower but still acceptable consistency among annotators are classified as hard difficulty, reflecting more subtle differences and greater challenge.

To further validate the grouping, we calculate Fleiss's Kappa coefficient for each level. The hard group exhibits moderate inter-annotator agreement ($0.4 < \kappa \le 0.6$), while the medium group shows high agreement ($\kappa > 0.6$).

Finally, we construct the benchmark data as shown in Table 2. For specific examples and concrete prompts been used please refer to Appendix H and Appendix I.

Table 2: The statics of PERSONAFEEDBACK

| | Specific | | | General | | | |
|---|---|---|---|---|---|---|---|
| **Easy** | **Medium** | **Hard** | **Easy** | **Medium** | **Hard** | **Total** |
| 1108 | 1667 | 1682 | 1510 | 1321 | 1010 | 8298 |

### 3.5 REWARD MODEL TRAINING

We construct a dataset consisting of 10,000 data points for reward model training. We use the responses generated by GPT-4o-mini given the persona profile as the chosen response and the responses generated without the persona as the rejected one to form training pairs. Then, we select Qwen2.5-0.5B-Instruct and Gemma-2B-it as our base models and adopt Bradley-Terry (BT) (Bradley & Terry, 1952) as the training objective. The mathematical formulation of the Bradley-Terry (BT) loss is as follows:

$$\mathbb{P}(a^1 \succ a^2 | x, a^1, a^2) = \frac{\exp(r^*(x, a^1))}{\exp(r^*(x, a^1)) + \exp(r^*(x, a^2))} = \sigma(r^*(x, a^1) - r^*(x, a^2)),$$

where $x$ represents the query, $a^1$ represents the chosen response, and $a^2$ represents the rejected response. Results and details in Appendix E show that such intuitive and simple preference data can effectively improve model performance on personalized benchmarks.

## 4 EXPERIMENTS

### 4.1 EVALUATION SETTINGS

We evaluate a wide range of models, including proprietary models and open-sourced models[2] in PERSONAFEEDBACK. This comprehensive evaluation aims to assess the performance of the reward models in personalized scenarios, as shown in Table 3.

We also explore models performance on three different settings in PERSONAFEEDBACK: (1) **Persona Profile:** The model has access to the configuration information of the persona. (2) **RAG:** The model do not maintain or explicitly learn the user's persona, but it can retrieve and use relevant user memory data to select more appropriate responses. (3) **No Persona:** The model does not have access to any personalized information. This setting serves as a baseline.

**Metric** We use Accuracy as the evaluation metric, measuring the model's ability to select the best personalized response and recognize high-quality answers under three levels of task difficulty. For classifier-based reward models, we define a correct classification when the model assigns a higher score to the chosen response than to the rejected response in the given persona $P$ and the input prompt $x$. For generative reward models, we use prompts to guide the model in selecting among the response options provided, thereby evaluating its ability to choose personalized content.

---

[2]See Appendix G for a complete list of models, including their references and Huggingface links.

Table 3: PERSONAFEEDBACK results for different model groups, including reasoning models, chat models, open-source models and reward models. Within each setting, models are sorted by **Total Avg.** in descending order. The highest score in each column is highlighted in **bold**.

| Model | Specific | | | | General | | | | Total Avg. |
|---|---|---|---|---|---|---|---|---|---|
| | Easy | Medium | Hard | Avg. | Easy | Medium | Hard | Avg. | |
| **Reasoning** | | | | | | | | | |
| o3-mini | 91.8 | **77.5** | 68.6 | **77.7** | 89.1 | 83.7 | 70.5 | 82.4 | 79.9 |
| o4-mini | 88.5 | 76.3 | 70.0 | 77.0 | 88.1 | 85.1 | 71.2 | 82.6 | 79.6 |
| Gemini-2.5-pro-exp-03-25 | 88.6 | 76.0 | 67.0 | 75.7 | **90.7** | 86.4 | **71.5** | **84.2** | 79.6 |
| Deepseek-R1 | 90.8 | 72.1 | **71.3** | 76.4 | 88.8 | **86.9** | 68.9 | 82.9 | 79.4 |
| o1-preview-2024-09-12 | **95.5** | 71.6 | 69.6 | 76.8 | 88.3 | 84.6 | 70.9 | 82.5 | 79.4 |
| Gemini-2.0-flash-thinking-exp | 89.8 | 76.3 | 68.0 | 76.5 | 89.7 | 82.4 | 69.8 | 82.0 | 79.0 |
| o1-mini | 88.3 | 76.4 | 67.5 | 76.0 | 88.2 | 86.0 | 68.9 | 82.4 | 79.0 |
| **Chat** | | | | | | | | | |
| GPT-4.1 | 90.6 | **76.5** | 69.1 | **77.2** | 89.0 | **85.7** | **71.1** | 83.2 | 80.0 |
| GPT-4.5-preview | 89.0 | 76.2 | 67.2 | 76.0 | **91.0** | 85.6 | 69.3 | **83.4** | 79.4 |
| Deepseek-V3 | 89.0 | 75.0 | 68.0 | 75.8 | 90.4 | 84.6 | 68.3 | 82.6 | 78.9 |
| GPT-4o-2024-11-20 | 88.8 | 76.2 | 68.6 | 76.5 | 89.2 | 84.5 | 66.5 | 81.6 | 78.9 |
| Claude-3-5-sonnet-20241022 | 89.2 | 73.6 | **70.2** | 76.2 | 89.4 | 84.1 | 67.5 | 81.8 | 78.8 |
| Claude-3-7-sonnet-20250219 | **90.7** | 72.8 | 64.8 | 74.2 | 89.0 | 83.1 | 70.1 | 82.0 | 77.8 |
| Gemini-2.0-flash | 89.3 | 73.7 | 64.6 | 74.1 | 88.4 | 84.3 | 69.0 | 81.9 | 77.7 |
| GPT-4o-mini | 87.3 | 73.4 | 63.3 | 73.0 | 87.2 | 84.4 | 68.8 | 81.4 | 76.9 |
| Claude-3-haiku-20240307 | 88.6 | 73.0 | 65.7 | 74.1 | 86.1 | 83.9 | 63.8 | 79.5 | 76.6 |
| GPT-4-turbo | 86.7 | 67.0 | 67.0 | 71.9 | 87.7 | 82.9 | 67.3 | 80.7 | 76.0 |
| **Open-Source** | | | | | | | | | |
| QwQ-32B | **88.8** | **74.7** | 66.6 | 75.1 | **89.3** | 84.7 | **68.1** | 82.1 | 78.3 |
| R1-Distill-Qwen-32B | 87.9 | 72.7 | 66.2 | 74.0 | **89.3** | **86.1** | 65.9 | 82.0 | 77.7 |
| Qwen2.5-32B-Instruct | 86.7 | 74.4 | **68.4** | **75.2** | 87.5 | 84.3 | 64.3 | 80.3 | 77.6 |
| R1-Distill-Qwen-14B | 87.2 | 73.7 | 62.9 | 73.0 | 87.0 | 81.4 | 66.9 | 79.8 | 76.1 |
| Qwen2.5-14B-Instruct | 84.0 | 72.0 | 66.3 | 72.8 | 86.4 | 83.3 | 65.1 | 79.7 | 76.0 |
| Qwen2.5-7B-Instruct | 81.4 | 63.4 | 62.0 | 67.3 | 83.9 | 79.3 | 60.7 | 76.2 | 71.4 |
| Llama-3-8B-Instruct | 71.0 | 59.8 | 53.2 | 60.1 | 50.9 | 51.0 | 50.6 | 50.9 | 55.8 |
| **Reward Model** | | | | | | | | | |
| INF-ORM-Llama3.1-70B | **85.2** | **75.9** | **70.2** | **76.1** | 88.1 | **85.2** | 69.7 | **82.3** | 79.0 |
| RM-Mistral-7B | 83.7 | 73.1 | 67.8 | 73.7 | 88.4 | 82.4 | **70.5** | 81.6 | 77.4 |
| LDL-Reward-Gemma-2-27B-v0.1 | 78.3 | 74.0 | 69.4 | 73.3 | 85.4 | 82.9 | 67.6 | 79.9 | 76.4 |
| Llama-3-OffsetBias-RM-8B | 83.5 | 69.8 | 67.2 | 72.2 | **88.8** | 79.4 | 65.3 | 79.4 | 75.5 |
| Skywork-Reward-Llama-3.1-8B | 74.7 | 72.2 | 67.6 | 70.3 | 82.5 | 81.3 | 68.1 | 78.3 | 74.4 |
| QRM-Llama3.1-8B-v2 | 76.8 | 71.2 | 65.2 | 70.3 | 79.1 | 76.5 | 62.5 | 73.8 | 71.9 |
| ArmoRM-Llama3-8B-v0.1 | 54.2 | 62.6 | 55.7 | 57.9 | 82.9 | 75.3 | 54.7 | 72.9 | 64.8 |

[*] All settings (easy, medium, hard) in the table above are binary choices. Therefore, the random baseline is 50.

## 4.2 MAIN RESULTS

> **Takeaway 1: Enhancing Reasoning Does NOT Yield To Better Personalization**
>
> In personalization QA, reasoning models fail to demonstrate competitive advantages over non-reasoning counterparts despite their enhanced reasoning capabilities.

As shown in Table 3, the average scores of reasoning models such as o3-mini and o4-mini on specific and general tasks are similar to those of chat models such as GPT-4.1 and GPT-4.5-preview. This indicates that enhancing reasoning ability alone does not necessarily translate into better personalization ability. Furthermore, in the comparison of open-source models, we observe that the performance of R1-Distill-Qwen-32B is lower than that of Qwen2.5-32B-Instruct. It is worth noting that even the top proprietary models have relatively low average accuracy on Hard difficulty tasks, suggesting that even the most advanced models still have room for improvement in understanding the subtlety of personalized responses.

> **Takeaway 2: Larger is Better**
>
> As the number of parameters increases, the performance of open-source models improves steadily, with large-scale models showing a clear advantage in handling personalized tasks.

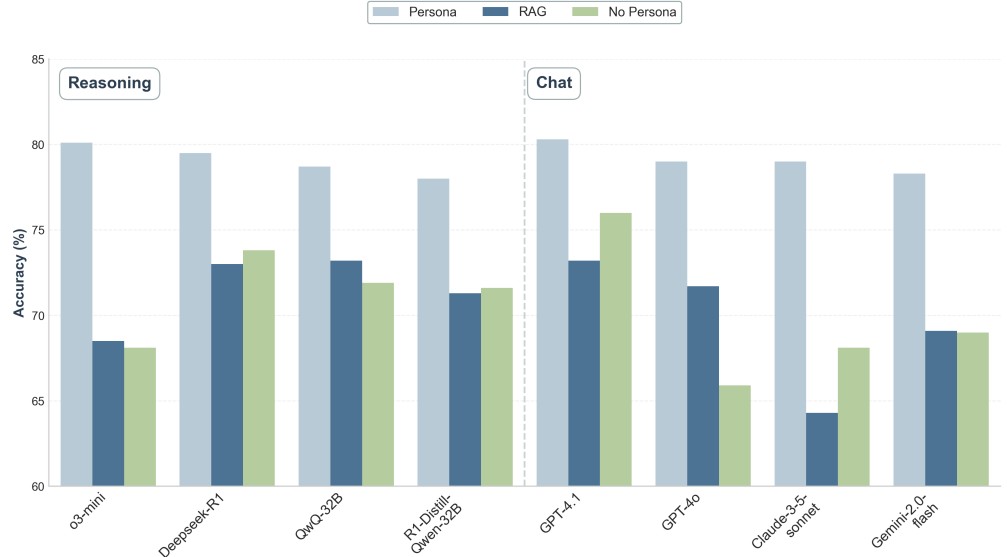

Figure 3: Comparison of Reasoning and Chat models under Persona, RAG, and No Persona settings.

As shown in Table 3, for the Qwen series, as the parameter scale increases from 7B to 32B, the model's performance improves steadily, with an 11-point increase in the Specific Medium task. The R1-Distill-Qwen series shows a similar trend. The average accuracy of QwQ-32B and Qwen2.5-32B-Instruct is comparable to that of some proprietary models such as GPT-4-turbo and 4o-mini.

**Takeaway 3: Reward Models Struggle with Personalized Questions**

Although SoTA reward models outperform base models on general questions, they lag behind on user-specific ones.

As shown in Table 3, models like INF-ORM-Llama3.1-70B (Minghao Yang, 2024)(ranks first in RewardBench Lambert et al. (2024)) achieve higher average scores on general questions than most generative models, likely due to their alignment with general question-answer pairs. However, for specific questions—especially in the easy and medium settings—reward models show limited generalization compared to generative models. This suggests that while current open-source general reward models are effective for general and potentially personalized queries, there is still room for improvement in handling more personalized questions.

**Takeaway 4: RAG Falls Short in Personalization**
**Takeaway 5: Persona Learning Should Be Made Explicit**

By comparing the RAG setting with the No Persona setting, we observed that LLMs do not benefit from the contextual information brought by the relevant content.

As shown in Figure 3, the Persona Profile-based setting consistently outperforms RAG and No Persona across all tasks. Surprisingly, RAG performs similarly to No Persona in most cases, which seems counterintuitive, as RAG should provide more relevant information than a model with no personalization. The main reasons may include: (1) In the RAG setting, the model needs to infer user preferences from the retrieved fragmented memories, which is an implicit reasoning task posing higher requirements of model's ability. On the other hand, the No Persona setting allows the model to focus on the quality of the answer to the question itself, without interference from potentially noisy memory data. (2) The retrieved information may contain noise or conflicting details. More experimental details about RAG can be found in the appendix F.

For example, when a user from Northeast China asks "What should I eat to recover faster after skiing?", the memory retrieved under the RAG setting only includes knowledge related to a fat-loss diet, but lacks crucial information about the user's origin in Northeast China. As a result, the model

cannot provide personalized advice tailored to the cold environment and regional dietary habits. This illustrates that personalized information should be explicitly learned and utilized, as relying solely on the retrieval mechanism cannot ensure the effective use of key information.

### 4.3 Correlation with Five Aspects of HelpSteer2

We used QRM-Llama3.1-8B-v2 (Dorka, 2024), a top-performing reward model on RewardBench, to estimate the five aspects proposed by HelpSteer2: (Wang et al., 2024d) (`helpfulness`, `correctness`, `coherence`, `complexity`, and `verbosity`). For each aspect, we computed the accuracy by checking whether the chosen response scored higher than the rejected response (e.g., if chosen.helpfulness > rejected.helpfulness, it is counted as correct in helpfulness). The results are reported in Table 4 as accuracy for each aspect over the whole set, as well as for the easy, medium, and hard subsets.

| Subset | Hel. | Cor. | Coh. | Comp. | Ver. |
|--------|------|------|------|-------|------|
| Easy   | 0.78 | 0.76 | 0.70 | 0.73  | 0.84 |
| Medium | 0.71 | 0.71 | 0.72 | 0.71  | 0.70 |
| Hard   | 0.65 | 0.65 | 0.65 | 0.61  | 0.61 |
| Whole  | 0.70 | 0.70 | 0.69 | 0.68  | 0.70 |
| Easy   | 0.78 | 0.78 | 0.76 | 0.74  | 0.82 |
| Medium | 0.75 | 0.76 | 0.76 | 0.70  | 0.69 |
| Hard   | 0.63 | 0.62 | 0.66 | 0.63  | 0.67 |
| Whole  | 0.73 | 0.73 | 0.73 | 0.70  | 0.74 |

Table 4: Accuracy on five HelpSteer2 aspects for the Specific (top) and General (bottom) sets.

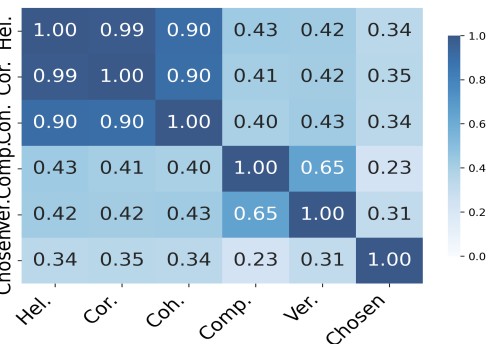

Figure 4: Correlation heatmap between the selection label and the five HelpSteer2 aspects.

**Observation on Accuracy:** The accuracy results in Table 4 provide several insights into the composition of our dataset. For the easy subset, where the comparison is often between a general response and a personalized one, the model achieves notably high accuracy on the verbosity aspect. However, as the difficulty increases in the medium and hard subsets, the accuracy drops across all five aspects. This trend indicates that our dataset is designed with a clear progression in difficulty, and as the questions become more difficult, it is not enough to rely on any single HelpSteer2 aspect. Instead, a higher level of personalization and helpfulness is required to distinguish answers in these harder cases. In addition, the accuracy values and trends are consistent across both the Specific and General sets, demonstrating the robustness of our data construction method.

**Analysis of Correlation:** Afterwards, we compiled a list of all responses with their scores on the five HelpSteer2 dimensions, along with a *chosen* flag (1 if the response was selected in the pairwise comparison, else 0). We then calculated the correlation coefficients (using Pearson (Freedman et al., 2007) correlation) between each dimension and the *chosen* label. As shown in Figure 4, the "personalization" metric shows little correlation from the five aspects addressed in Helpsteer2. This suggests that our personalization metric captures information not covered by existing dimensions, highlighting its distinctiveness and importance.

## 5 Conclusion

In this work, we introduce PERSONAFEEDBACK, a novel benchmark that decouples persona inference from personalization, enabling a focused evaluation of how well LLMs can adapt their responses to explicit user personas. We curate a dataset of 8,298 **human-annotated** test cases, categorized into easy, medium, and hard tiers, providing a comprehensive evaluation suite for LLM personalization. We conduct extensive evaluation across a wide range of models, revealing critical insights into the limitations of current systems, such as the insufficient effectiveness of long-reasoning models, the advantage of larger models, and the challenges faced by reward models and retrieval-augmented frameworks in personalization tasks. Our benchmark, along with detailed annotation protocols and evaluation tools, is made publicly available to support future research into personalized adaptation in LLMs.

## 6 REPRODUCIBILITY STATEMENT

We ensure reproducibility: data, models and evaluation settings are detailed in Section 3, Appendix G and Appendix F; training parameters and public libraries evolved are in Appendix E; the dataset has been released.

## 7 ETHICS STATEMENT

This research adheres to ICLR's Code of Ethics. Datasets are public/synthetic with no real user info, thus avoiding privacy risks. Human evaluators provided informed consent with anonymity guaranteed.

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

## A  MORE DETAILS ABOUT THE STATISTICS

In designing the PersonaFeedback benchmark, we carefully considered scenario diversity and constructed the dataset based on real-world statistical distributions. Our data covers a broad range of domains, including daily life, entertainment, productivity, healthcare, and financial advice, with the specific distribution shown in the table 5.

Table 5: The Distribution of Scenario Statistics.

| Category | Percentage (%) | Included Scenarios |
|---|---|---|
| Recommendation Systems | 23.47 | Shopping Advice, Hotel & Restaurant Recommendations, Social Activity Suggestions, Outfit Matching, Investment Advice, Hairstyle Suggestions, Personalized Playlists, Interior Design Advice, Furniture Arrangement, Sustainable Shopping |
| Entertainment & Leisure | 20.79 | Music, Travel Tips, Trip Planning, Trending Rankings |
| Education & Learning | 19.02 | Knowledge Expansion, Grammar Practice, Exam Review, Speaking Practice, Interview Preparation |
| Office & Productivity | 18.79 | Schedule Management, Document Editing, Meeting Scheduling, Project Management, Email Writing, Resume Optimization, Material Selection |
| Health & Medical | 10.53 | Nutrition Information Query, Diet Planning, Symptom Checker, Medication Information, Fitness Tracking, Skincare, Pet Health |
| Personal Growth | 4.24 | Interpersonal Relationships, Career Advice, Listening & Support, Self Reflection |
| Technical Development | 2.02 | Code Debugging, Code Optimization |
| Finance | 1.10 | Financial Analysis, Budget Management, Carbon Footprint Calculator |

Below is a complete example of a user persona profile.

---

**Persona Profile**

### Demographic Info

**Name:** Brandon
**Age:** 26
**Gender:** Male
**Nationality:** Chinese
**Languages:** Chinese, English
**Education:** Master's student in Computer Vision
**Personality:** Introversion(I), Intuition(N), Thinking(T), Judging(J)
**Values and Interests:**
Passionate about machine learning, particularly focused on computer vision.
Advocates for liberal values.
Actively supports open-source projects and community building.
Enthusiastic about gaming and esports.

### DLifestyle Preferences:

**Food & Drink:**
Typically eats lunch and dinner at the school cafeteria or near the lab on weekdays.
Occasionally explores refined restaurants or cafés with classmates.
Enjoys ordering late-night snacks when working overtime.

**Entertainment & Shopping:**
Regularly visits cinemas on weekends or during free time.
Likes relaxing with friends at board game cafés.

**Daily Commute:**
Occasionally enjoys walks in city parks during weekends to relax.

---

**Persona Profile**

**Travel Habits**
**Business Travel:**
Occasionally travels to major tech hubs for academic conferences or tech exhibitions.
Likes sampling local specialties during free time on business trips.

**Leisure Travel:**
Prefers family trips during winter holidays, exploring natural landscapes.
Always prepares detailed travel guides and books accommodations in advance.

**Social Life:**
Highly active on GitHub, regularly releasing and maintaining open-source projects.
Follows esports events and cutting-edge tech developments on social platforms.
Frequently plays badminton with classmates in the afternoons or weekends.

**Productivity Tools Study & Work:**
Uses Zotero for literature reading and management.
Prefers Office Suite or Markdown tools to record notes, organize project planning, and document requirements.

**Fitness Routine:**
Uses Keep fitness app to maintain healthy living habits.

**Content Interests:**
Enjoys watching League of Legends (LOL) esports matches.
Has a strong interest in electronic products, staying updated with new products and technology trends.

## B  PERSONA SELECTION PROTOCOL

**Selection Process**  To ensure that the personas we build are authentic and diverse, we implemented two rounds of quality screening for the initial 1,700 personas.

1. **Round 1: Authenticity and Consistency**
   - Exclude personas whose professions, behaviors, or lifestyles are disconnected from modern society.
   - Exclude overly idealized or internally contradictory personas.
   - Exclude personas that are overly dramatized or reliant on stereotypes.
2. **Round 2: Diversity and Balance**
   - Avoid over-concentration in specific profession categories.
   - Avoid homogeneity in interests, habits, and other dimensions.

## C  QUESTION FILTERING PROTOCOL

After generating the set of questions, we apply the following manual filtering criterion to ensure the quality and naturalness of the final questions:

- **Logical Inconsistency**: Filtered out questions that were logically inconsistent or forced unrelated topics together.
- **Unnatural Phrasing**: Filtered out questions that were awkwardly phrased or unnatural.

---

**Examples of Filtered Questions**

**Question1:**
I am currently optimizing the document processing process. Is there a way to improve housework efficiency at the same time ?

**Reasons for filtering:**
This question awkwardly combines document processing workflows and house efficiency, two unrelated domains that lack natural coherence.

**Question2:**
How can gardening knowledge be used to optimize software architecture ?

**Reasons for filtering:**
The question bridges fields that are too unrelated, which goes against conventional logic.

---

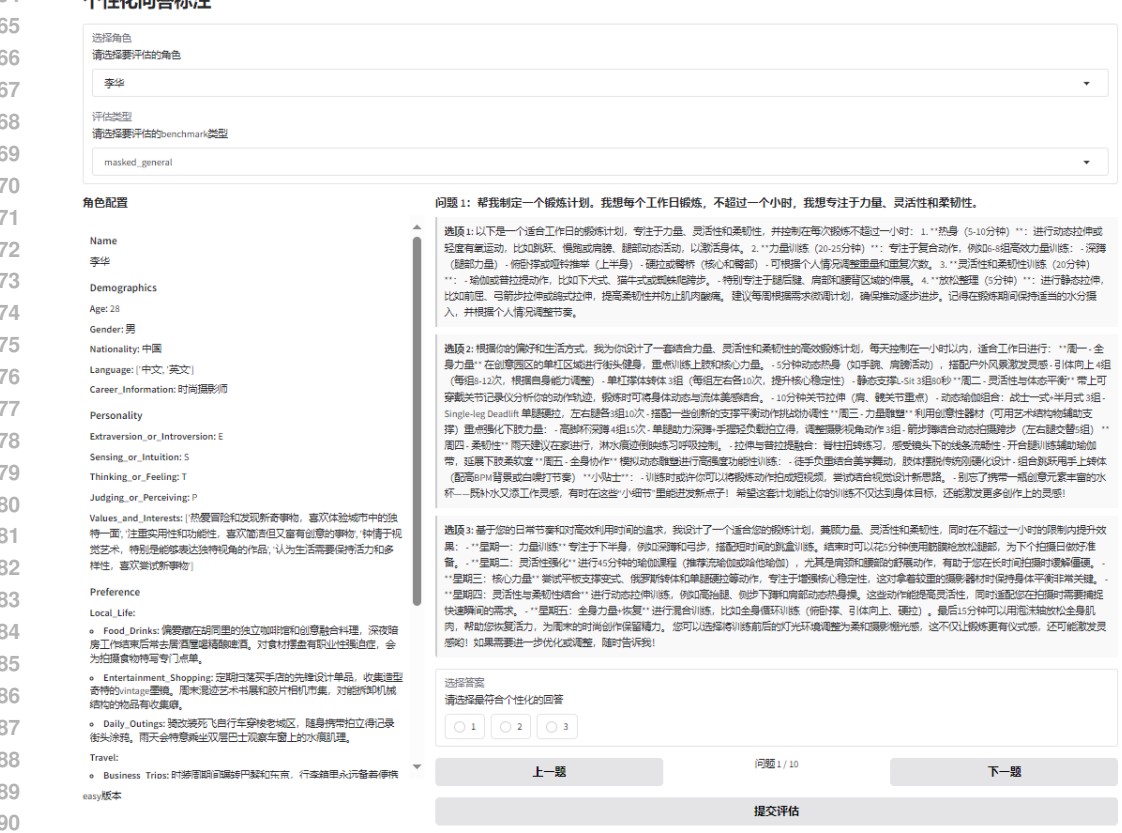

Figure 5: A screenshot of the human annotation.

# D ANSWER SELECTION PROTOCOL

**Task Description** The human evaluator needs to simulate the user based on the persona profile provided and select the most appropriate answer based on the following criteria:

**Selection Criteria**

- **Helpfulness** The answer should precisely address the user's specific question, rather than being a generic response or a forced combination of information from the user configuration that results in a strained answer.
- **Personalization** The answer should reflect attention to multiple fields of information in the user's configuration and appropriately integrate relevant aspects to provide personalized assistance.

# E    REWARD MODEL TRAIN DETAILS AND RESULTS

Table 6: PERSONAFEEDBACK train results for reward model models.

| Model | Specific | | | | General | | | | |
| | Easy | Medium | Hard | Avg. | Easy | Medium | Hard | Avg. | Total Avg. |
|---|---|---|---|---|---|---|---|---|---|
| Qwen2.5-0.5B-Instruct(generative) | 49.8 | 51.3 | 50.5 | 50.6 | 49.8 | 50.8 | 49.7 | 50.1 | 50.4 |
| +HelpSteer2 Trained | 33.6 | 61.6 | 54.7 | 52.0 | 39.7 | 52.5 | 51.5 | 47.2 | 49.8 |
| +HelpSteer2 Personalized Trained | 85.9 | 68.3 | 57.1 | 68.4 | 79.1 | 71.9 | 57.4 | 70.9 | 69.6 |
| Qwen2.5-3B-Instruct(generative) | 70.1 | 64.2 | 57.6 | 63.2 | 70.7 | 71.0 | 55.8 | 66.9 | 64.9 |
| +HelpSteer2 Trained | 59.6 | 64.3 | 62.7 | 62.5 | 74.5 | 73.4 | 57.2 | 69.6 | 65.8 |
| +HelpSteer2 Personalized Trained | 88.7 | 71.2 | 64.8 | 73.1 | 84.8 | 77.4 | 57.4 | 75.1 | 74.0 |
| Gemma-2b-it(generative) | 51.1 | 47.9 | 50.7 | 49.8 | 51.0 | 51.6 | 49.9 | 50.9 | 50.3 |
| +HelpSteer2 Trained | 71.7 | 64.2 | 57.3 | 63.5 | 73.4 | 71.2 | 60.0 | 69.1 | 66.1 |
| +HelpSteer2 Personalized Trained | 88.5 | 71.5 | 61.0 | 71.8 | 81.5 | 73.1 | 61.7 | 73.4 | 72.5 |

[*] All settings (easy, medium, hard) in the table above are binary choices. Therefore, the random baseline is 50.

Table 6 shows that base generative models (Qwen2.5-0.5B-Instruct and Gemma-2B-it) perform near random on personalized tasks, indicating that too small models cannot handle personalization out of the box, but Qwen2.5-3B-Instruct can get a score of 65 initially.

To enhance model performance, we selected 3632 pairs of data from Helpsteer2, These samples were selected based on a criterion where the chosen response demonstrated a helpfulness score at least 2 points higher than its rejected counterpart.

Subsequently, we combined this helpfulness-oriented data with our custom dataset of 10,000 preference pairs, utilizing the RLHFlow/RLHF-Reward-Modeling framework (Dong et al., 2024). The resulting BT reward models (shown in the "+HelpSteer2 Personalized Trained" rows) achieved substantial improvements across both specific and general question categories.

While simple, our preference data effectively improves model performance on PERSONAFEEDBACK.

Training details are as follows:

- For the "+HelpSteer2 Trained" setting, we use a maximum sequence length of 4096, a learning rate of 1e-5, and a batch size of 32, and train for one epoch.

  In the "+HelpSteer2 Personalized Trained" setting, both models use a maximum sequence length of 4096, a learning rate of 1e-5, a batch size of 64, and 1 epoch.

- We observe that varying the batch size between 64 and 256, or adjusting the learning rate between 1e-5 and 2e-5, has little impact on the results. Similarly, the choice of adjacent checkpoints has little effect on model performance.

# F    DETAILS ABOUT RAG

Our data collection process was not specifically tailored for RAG evaluation, but rather designed to simulate real-world scenarios (such as the Google ecosystem), where user personas are inferred from search history and content consumption behaviors, etc. This design means that, when applying a simple RAG approach (i.e., retrieving memories directly based on the query), a certain amount of irrelevant or weakly related memory information can be introduced, which challenges the effectiveness of the current RAG system on personalization tasks.

In such realistic settings, simple RAG (query-based memory retrieval) often introduces more noise than signal. This explains why RAG can sometimes underperform even the No Persona baseline, highlighting its current inadequacy for personalization.

To further examine this issue, we conducted an additional supplementary experiment as shown in table: building on the original simple memory retrieval, we introduced an LLM to filter the retrieved content. By explicitly providing user information, the LLM filters out irrelevant or weakly related memory fragments, ensuring that only high-quality and highly relevant memories are passed to the model for answer selection.

The improved RAG approach achieved higher accuracy on the benchmark than the original simple RAG, but still fell short of the results obtained by directly providing the full persona information.

Table 7: Comparison of Reasoning and Chat models under Persona, RAG, and No Persona settings.

| Model | Persona | RAG | No Persona |
|---|---|---|---|
| **Reasoning** | | | |
| o3-mini | 80.1 | 72.5 | 68.1 |
| Deepseek-R1 | 79.5 | 75.0 | 73.8 |
| QwQ-32B | 78.7 | 75.2 | 71.9 |
| R1-Distill-Qwen-32B | 78.0 | 74.3 | 71.6 |
| **Chat** | | | |
| GPT-4.1 | 80.3 | 76.2 | 76.0 |
| GPT-4o | 79.0 | 73.7 | 65.9 |
| Claude-3-5-sonnet | 79.0 | 70.3 | 68.1 |
| Gemini-2.0-flash | 78.3 | 73.1 | 69.0 |

Table 7 shows that the results further illustrate the limitations of simple RAG methods in personalization tasks and strongly validate our core conclusion: explicitly providing user persona information is both necessary and effective for improving personalization performance.

## G  LIST OF MODELS USED

Table 8: List of all models used in our experiments, with Hugging Face links where available.

| Model Name | Hugging Face Link |
|---|---|
| QwQ-32B (Team, 2025) | https://huggingface.co/Qwen/QwQ-32B |
| R1-Distill-Qwen-32B (DeepSeek-AI et al., 2025) | https://huggingface.co/deepseek-ai/DeepSeek-R1-Distill-Qwen-32B |
| Qwen2.5-32B-Instruct (Yang et al., 2024a) | https://huggingface.co/Qwen/Qwen2.5-32B-Instruct |
| R1-Distill-Qwen-14B | https://huggingface.co/deepseek-ai/DeepSeek-R1-Distill-Qwen-14B |
| Qwen2.5-14B-Instruct | https://huggingface.co/Qwen/Qwen2.5-14B-Instruct |
| Qwen2.5-7B-Instruct | https://huggingface.co/Qwen/Qwen2.5-7B-Instruct |
| Llama-3-8B-Instruct (AI@Meta, 2024) | https://huggingface.co/meta-llama/Meta-Llama-3-8B-Instruct |
| INF-ORM-Llama3.1-70B (Minghao Yang, 2024) | https://huggingface.co/infly/INF-ORM-Llama3.1-70B |
| RM-Mistral-7B (Xiong et al., 2024) | https://huggingface.co/weqweasdas/RM-Mistral-7B |
| LDL-Reward-Gemma-2-27B-v0.1 | https://huggingface.co/ShikaiChen/LDL-Reward-Gemma-2-27B-v0.1 |
| Llama-3-OffsetBias-RM-8B (Park et al., 2024) | https://huggingface.co/NCSOFT/Llama-3-OffsetBias-RM-8B |
| Skywork-Reward-Llama-3.1-8B (Liu et al., 2024b) | https://huggingface.co/Skywork/Skywork-Reward-Llama-3.1-8B |
| QRM-Llama3.1-8B-v2 (Dorka, 2024) | https://huggingface.co/nicolinho/QRM-Llama3.1-8B-v2 |
| ArmoRM-Llama3-8B-v0.1 (Wang et al., 2024a) | https://huggingface.co/RLHFlow/ArmoRM-Llama3-8B-v0.1 |
| Qwen2.5-0.5-Instruct | https://huggingface.co/Qwen/Qwen2.5-0.5B-Instruct |
| Gemma-2b-it | https://huggingface.co/google/gemma-2b-it |

# H EXAMPLES

Table 9: Examples of three levels of difficulty (easy, medium, hard).

| Query | Type | Response |
|---|---|---|
| If I want to focus on my health, what aspects do I need to consider? | Easy | **chosen:** Every morning when you exercise in the park, you can do some proper warm-up to prevent injuries... Sitting for long periods while working can easily lead to fatigue. It is recommended that you stand up and move your shoulders and neck every 45 minutes...
**reject:** Every morning, you can take 5 to 10 minutes to plan your day, ensuring that you prioritize important and urgent tasks before addressing non-urgent matters... |
| How can I uplift myself when I'm feeling down? | Medium | **chosen:** You can try flipping through some philosophy books you found in old bookstores on Fuzhou Road, such as "The Tibetan Book of the Dead." Such reading often provides emotional comfort and a deeper sense of meaning in life. If the weather outside is nice, you could also take a walk in the Guangfulin Ruins Park, where you can quietly observe and record the interactions between different people. This can help you discover simple and beautiful moments in life...
**reject:** When feeling down, you might try browsing through some philosophical reading materials, such as related books or personal reflection notes from others. These words may bring a certain spiritual strength... |
| How to stay healthy amidst a busy schedule? | Hard | **chosen:** 1.Morning Planning: Continue to stick to the 5:30 am high-intensity interval training and incorporate health goals into your schedule using data from your Apple Watch.... 2. Healthy Eating: During work hours, choose your usual cold brew coffee paired with a healthy light meal, such as organic salad and protein-rich sides. You might consider pre-ordering high-quality takeout for the week to reduce waiting time while ensuring the freshness and nutrition of the ingredients... 3. Weekend Recovery: Use weekends to try your favorite Spanish tapas restaurant and combine it with light hiking or cultural exploration in the area to rejuvenate both body and mind...
**reject:** You can combine your morning running routine with exploring a high-quality café near the hotel that offers healthy breakfast options. This not only energizes you but also adds a sense of enjoyment to your workday. Make use of the space in the airport lounge to choose nutritious light snacks and fruits instead of fast food. Additionally, take advantage of fragmented time in the mornings or evenings to do some low-intensity stretching or meditation to help your body recover. Consider portable smart health devices (such as a portable heart rate monitor or a yoga practice guide) that align with your preference for efficient tools; these can easily fit into your work or travel environment while enhancing your health management efficiency. |

Table 10: Examples of different types.

| Query | Type | Response |
|---|---|---|
| What methods can be used to better record the details during a trip? | Uncomfortable | **chosen:** You could try combining recording with creative expression. It's recommended to carry a lightweight film camera to capture delicate street scenes and cultural moments, such as a steaming bowl in an old teahouse or the handwritten signs of a flower shop on the corner. At the same time, keep a Moleskine sketchbook handy to jot down inspirations and observations, capturing those unique moments... 
 **reject:** You can carry a lightweight film camera to capture changes in light and shadow, such as the dappled sunlight on bamboo chairs in the street or the reflections outside a café window in an alley... These methods not only align with Chen Jialing's pursuit of non-commercial recording but also help capture more delicate moments... Chen Jialing can also use her fragrance collection bottle to gather the scents from markets, cafés, or guesthouses as another form of sensory recording... |
| Are there any films that provide a sense of thrilling adventure while also featuring creative action sequences? | Over-personalization | **chosen:** If you want to experience the thrill of adventure while enjoying creatively packed action sequences, I highly recommend the "Agent Extreme" series. This series is filled with extreme sports elements such as skydiving and downhill skateboarding, guaranteed to ignite your spirit of adventure... After watching, if you're interested, you can try to mimic some of the moves from the film and even use a GoPro to capture your own creative extreme sports clips... 
 **reject:** The "Agent Extreme" series hits all the right notes for you—the scene where the main character rides a modified skateboard and dives down a snowy mountain is so thrilling that it practically defies Newton's laws; the collision of physics and extreme sports will definitely get your DNA racing. I recommend paying special attention to the part where Vin Diesel descends the steps of a bridge; you'll surely find yourself jotting down three pages of notes on ideas for upgrading your fixie bike... By the way, the parabolic trajectory of the motorcycle surfing at the end of the film is definitely worth writing next to your game sketches in your physics notebook as a fun bonus... |

# I PROMPTS USED IN PERSONAFEEDBACK DATA GENERATION

---

**Prompt for generating questions**

**System Prompt:**
You are a professional role-player. Based on the following character information, you need to generate one natural question that the character would ask an AI assistant.

**Important rules:**
1. The question must be based on a scene. Avoid purely factual queries. The question should focus on open-ended, exploratory, or subjective judgment types.
2. The question must sound natural, self-consistent and realistic, as if asked by a real person. Do not awkwardly mash memory scenes and persona configuration. You can reference the persona configuration for inspiration, but the question should arise organically from the scene.

Below is the persona config:
{persona_description}
Below is the list of memory scenes:
{scenes}
Output format example:
{example}

---

**Prompt for naturally phrased questions**

**System Prompt:**
Your task is to determine whether the following question is natural, coherent, and aligns with what a user would typically ask an AI assistant. If you find any logical contradictions in the question, you should rewrite it to remove the unreasonable parts and rephrase it into a natural question.

Please format your output as follows:
<conclusion >
Answer whether to rewrite here, output True or False (True means it needs to be rewritten)
</conclusion >
<question >
Output the rewritten question based on the conclusion of the conclusion. If the conclusion is True, output the rewritten content. if it is False, output the original question.
</question >
The following is an example of an unnatural question that needs to be rewritten:
{example}
The following is the question:
{question}

---

---

**Prompt for generating answers**

**System Prompt:**
You need to simulate responses from an answer the user's questions as the AI assistant based on the provided user configuration.
You need to simulate responses from an AI assistant to user questions. Answer user questions as the AI assistant based on the provided user configuration.
First, consider helpfulness: the answer should be more precise in addressing the user's specific question. Eliminate unnatural or redundant phrases, ensuring that the answer is natural and fluent.
Next, consider personalization: you can reference relevant personalized fields, but do not insert them awkwardly.

**Important rules:**
1. The response should reflect the tone and style, the professional AI assistant, friendly and helpful.
2. Take into account the user's preferences and characteristics to provide a personalized response.
3. The answer should meet the user's needs without overtly showcasing field information. Instead, infer what the user might want based on their existing persona configuration, rather than just combining field details.
4. The response should directly solve the user's question while reflecting an understanding of their preferences.

Here is the current persona config:
{persona_description}
Here is an example of output:
{example}
Here is the user question:
{question}
Generate a personalized answer:

---

**Prompt for answer improvement**

**System Prompt:**
You are a professional AI assistant who specializes in improving the quality of responses. You need to improve an existing answer to make it more personalized and helpful.
Original question:
{question}
The answer to be improved:
{original_answer}

Please improve the response:
The answer should more accurately address the user's specific question, eliminating any unnatural or redundant parts in the original response. Make sure that the answer flows naturally without awkwardly inserting any information from the user's configuration.
Put the improved response within <improved_answer >tags. The revised answer should be noticeably more natural than the original one. Ensure that the word count does not increase or decrease!

Output format:
<reasoning >
The changes to the answer are output here.
</reasoning >
<improved_answer >
The modified answer content.
</improved_answer >

---

**Prompt for answer degradation**

**System Prompt:**
You are an expert in modify the AI assistant answer. You are required to modify an existing answer to make it less personalized or less helpful.
Original question:
{question}
The answer to be degraded:
{original_answer}

Please modify the previous answer by choosing one of the following directions to lower its quality:
1. Reduce personalization: Make the answer more generic. For example, change a personalized element in the original response into something more universal, something that would be acceptable to anyone, or at least not fully aligned with the user's specific interests and preferences.
2. Lower helpfulness: Make the answer vague or less precise, not solving the user's specific problem well. For instance, add overly general or irrelevant information, or provide vague or overly broad advice.

Place the degraded answer in the <degraded_answer >tag. The modified answer should be more generic and less helpful than the original, but still maintain basic logic and structure. Ensure that the word count does not increase or decrease!
Output format:
<reasoning >
The changes to the answer are output here.
</reasoning >
<degraded_answer >
The modified answer content.
</degraded_answer >

---

## J  LIMIATIONS

Although PERSONAFEEDBACK significantly advances the research on personalized evaluation of large language models (LLMs), some limitations still exist. The binary choice evaluation method we use effectively quantifies differences in personalization capabilities, but human evaluators' judgments are inevitably influenced by subjective factors, especially in more challenging cases. Moreover, despite our efforts to construct diverse and realistic user personas, the created characters may still contain certain biases or simplifications, failing to fully capture the complexity and nuances of real users.

## K  THE USE OF LARGE LANGUAGE MODELS (LLMS)

Large Language Models (LLMs) support the research process in two respects: polishing the writing and providing guidance on LaTeX operations.

