# OpenReview forum: "PERSONAFEEDBACK: A Large-scale Human-annotated Benchmark For Personalization"
_ICLR.cc/2026/Conference — Submitted to ICLR 2026_

### Official Review · Reviewer_SEns · 2025-10-24

**Soundness:** 2
**Presentation:** 3
**Contribution:** 2
**Rating:** 2
**Confidence:** 3

**Summary:**

This paper introduces the PersonaFeedback dataset for evaluating personalization. It consistens of 8298 test cases that are grouped in easy, medium, and hard. The authors find that, with this dataset, enhances reasoning does not improve personalization and RAG also does not help. However, larger model do perform better

**Strengths:**

The authors are addressing an important gap in the field of needing high quality dataset to evaluate personalization.

The authors have interesting results showing that RAG is not helpful for personalization but that larger models do tend to do better.

**Weaknesses:**

Each section of the methods could be better motivated. It wasnt always clear to me what the point of each section was.

The paper could benefit from a more detailed description how the annotators were selected and the instructions they were given as this can have a large influence on the resulting dataset.

The paper could also benefit from a limitations section describing the limitations of the dataset and what it would not be as useful for evaluating

It is not clear what the reward training model is?

My biggest concern is that the paper is over-claiming what the dataset enables. The authors state that PersonaFeedback is: "a new benchmark that directly evaluates LLMs’ ability to provide personalized responses." However, from my understanding the dataset is evaluating where the LLM is able to choose the most personalized response from a set of options, not whether it can actually generate a personalized response. these are two very different things.

**Questions:**

How were the 20 intial seeds collected?

how well does this generalize across cultures? Do the personas tend to be specific to the Chinese culture or is there a mix?

What is the difference between specific and general questions?

Is it possible that for the difficult level there could just be several answers that could be considered correct?

---

> ### Author Response · Authors · 2025-11-17
>
> ### Weaknesses:
>
> > Q1: The paper could benefit from a more detailed description how the annotators were selected and the instructions they were given as this can have a large influence on the resulting dataset.
>
> Yes, this is an important point. The quality of our annotators and the guidelines they followed have already been described in detail in ​**Appendix D ​**​(ANSWER SELECTION PROTOCOL) of the paper, and our annotation interface is presented in Figure 5. As we mentioned in **Section 3.4** of the main text, we hired nine human evaluators. In Appendix D, we elaborate on their core task: first, to carefully read and role-play the given Persona Profile. Subsequently, they were required to select the more suitable response from a pair of answers based on two strict and clear criteria. These criteria are:
>
>
>
> * ​**Helpfulness**​: The answer must precisely address the user's specific problem, rather than being a generic and irrelevant response.
>
> * ​**Personalization**​: The answer must reflect attention to multiple information fields in the user's profile and appropriately integrate relevant aspects (such as interests, occupation, preferences, etc.) into the reply, rather than simply stacking information awkwardly.
>
>
>
> > Q2: The paper could also benefit from a limitations section describing the limitations of the dataset and what it would not be as useful for evaluating
>
>
>
> We agree that a dedicated "Limitations" section is essential for the completeness of our paper. We will add this section in the revised version and will at least cover the following two aspects:
>
>
>
> 1. ​**Cultural Scope**​: We acknowledge that the Persona Profiles in our current dataset are primarily constructed based on Chinese and mainstream Western cultural backgrounds. Consequently, the benchmark may have limitations in evaluating a model's ability to personalize responses for other diverse cultural contexts.
>
> 2. ​**Long-term Validity**​: The benchmark is established based on the evaluation of models prevalent in early 2025. We recognize that if future large language models undergo fundamental shifts in response style, interaction patterns, or quality, leading to output distributions that differ significantly from current models, our benchmark might require an update to remain an effective tool for evaluating these new capabilities.
>
>
>
> We believe that the inclusion of this discussion will make our paper more rigorous and comprehensive.
>
>
>
> > Q3: It is not clear what the reward training model is?
>
>
>
> Not sure which one you want to ask about, regarding 'reward model', we guess you might be asking about one of these two.
>
>
>
> 1. **SOTA Reward Models Used for Evaluation (see Table 3).** The models listed under the "Reward Model" category in Table 3 of our paper (e.g., INF-ORM-Llama3.1-70B, RM-Mistral-7B, LDL-Reward-Gemma-2-27B-v0.1) are existing, open-source, state-of-the-art (SOTA) general-purpose reward models. **We did not train these models.** We included them in our evaluation to benchmark how current SOTA general-purpose reward models perform on our personalization benchmark, PERSONAFEEDBACK. We found that despite their excellent performance on general benchmarks like RewardBench, their performance on our specific questions was not superior to generative models. This finding underscores the necessity of a dedicated benchmark for personalization.
>
> 2. **Experimental Models We Trained (see Section 3.5 and Appendix E).** As described in Section 3.5, we selected two open-source base models: Qwen2.5-0.5B-Instruct and Gemma-2B-it. The purpose of this training was ​**not to create a new SOTA reward model**​, but rather to demonstrate that our dataset of 10,000 preference pairs is effective and can be successfully used to train models to improve their ability to recognize personalized preferences. Using a Bradley-Terry (BT) loss function, we fine-tuned these two base models on our 10,000 (chosen/rejected) preference pairs. The results (shown in Appendix E, Table 6) indicate that after training, both Qwen2.5-0.5B-Instruct and Gemma-2B-it achieved significantly higher scores on PERSONAFEEDBACK (e.g., Qwen-0.5B's overall score increased from 50.4 to 69.6) compared to their original base models, thus validating the effectiveness of our dataset and methodology.

---

> > ### Author Response · Authors · 2025-11-17
> >
> > > Q4: My biggest concern is that the paper is over-claiming what the dataset enables. The authors state that PersonaFeedback is: "a new benchmark that directly evaluates LLMs’ ability to provide personalized responses." However, from my understanding the dataset is evaluating where the LLM is able to choose the most personalized response from a set of options, not whether it can actually generate a personalized response. these are two very different things.
> >
> >
> >
> > Thank you for this insightful comment. We completely agree that evaluating a model's ability to **choose** a personalized response (a discriminative task) is different from evaluating its ability to **generate** one (a generative task). We acknowledge that the core paradigm of PersonaFeedback is indeed discriminative, and this was a deliberate design choice.
> >
> >
> >
> > Evaluating free-form generation, such as with LLM-as-a-judge, is often subjective, hard to reproduce, and suffers from inconsistent standards, especially for open-ended questions. In contrast, a discriminative, pairwise comparison task provides an **objective, quantifiable, and stable** method to directly assess a model's understanding of the nuanced concept of "personalization," avoiding the ambiguity of generative evaluation.
> >
> >
> >
> > Furthermore, in real-world applications where generative models evolve rapidly, having a stable discriminator trained on human preferences is crucial. Such a model provides a reliable metric to consistently evaluate and compare the personalization quality of different generative models over time, without needing to constantly redesign the evaluation protocol. PersonaFeedback is designed precisely to fill this gap: it serves as the essential benchmark to validate the effectiveness and accuracy of these vital discriminators, ensuring that the evaluation of personalization is itself reliable and consistent.
> >
> >
> >
> > ### Questions:
> >
> >
> >
> > > Q1: How were the 20 initial seeds collected?
> >
> >
> >
> > Our specific process was as follows: First, we defined a comprehensive ​**Persona Schema**​, as exemplified by the "Brandon" profile shown in Appendix A. This template covers multiple dimensions, including demographics, MBTI, values, interests, and detailed lifestyle preferences (such as diet, entertainment, travel, social habits, work tools, etc.).
> >
> >
> >
> > Next, we recruited 20 volunteers from diverse backgrounds. We ensured that these volunteers were broadly representative in terms of profession (e.g., covering fields like technology, arts, education, and healthcare), age, interests, and life experiences to guarantee diversity in our initial seed pool. Finally, we asked these volunteers to fill out the Persona Schema based on their own real-life situations.
> >
> >
> >
> > > Q2: how well does this generalize across cultures? Do the personas tend to be specific to the Chinese culture or is there a mix?
> >
> >
> >
> > We acknowledge that while we strived to enhance the diversity of our personas by including a wide range of professions, interests, and lifestyles (see Sec. 3.1 ), the current dataset may be limited in its cultural scope due to constraints in our data sources and scope. The profiles may primarily reflect Chinese and mainstream Western cultural contexts, and we have not yet systematically incorporated perspectives from other global cultures, such as those from the Middle East, Latin America, or Africa.
> >
> >
> >
> > We fully agree with this limitation. In response to your valuable feedback, we will add**​ ​**a**​ discussion to the "Limitations" section** of our paper.
> >
> >
> >
> > However, we wish to emphasize that the **core methodology** we present for data construction and evaluation is culture-agnostic. We are confident that our framework can be readily applied to construct similar, high-quality personalization benchmarks for any specific cultural context, provided that culturally relevant seed data is available.

---

> > > ### Author Response · Authors · 2025-11-17
> > >
> > > > Q3: What is the difference between specific and general questions?
> > >
> > >
> > >
> > > This is a core distinction in our benchmark design, which we elaborate on in ​**Section 5.1 (Training Datasets)**​. The two question sets are designed to evaluate a model's personalization capabilities from two different and complementary perspectives.
> > >
> > >
> > >
> > > **Specific Questions** are questions that are ​**explicitly tailored to a given persona**​. These questions are dynamically generated. The process begins by inferring a user's traits (Pi) from their memory data. These traits are then combined with a random scenario (​*S*​) to produce a highly personalized query, following the formulation Qi=f(Pi,S). As a result, the question itself contains the persona's unique background (e.g., a fitness coach who enjoys Cantonese cuisine might ask, ​*"What are some light, high-protein Cantonese recipes you'd recommend after an intense workout?"*​). This category of questions tests the model's ability to accurately respond to the **explicit personalization cues** already present in a context-rich query.
> > >
> > >
> > >
> > > In contrast, **General Questions** are queries that anyone might ask. These are sampled from a large, public dataset (ShareGPT) and are filtered to retain only subjective, open-ended questions (e.g., ​*"How can I relax?"*​). This category tests a different skill: the model's ability to **proactively and correctly inject** the persona information it has been conditioned on, thereby transforming a generic answer into a personalized one. For instance, in response to the general query about relaxation, the model might suggest that the aforementioned fitness coach could relax by researching new recipes, rather than giving a generic recommendation like watching a movie.
> > >
> > >
> > >
> > > By including both of these dimensions—testing the model's ability to respond to **explicit cues** and its ability to inject ​**implicit knowledge**​—we believe our evaluation framework provides a more comprehensive and robust assessment of a model's true personalization proficiency.
> > >
> > >
> > >
> > > > Q4: Is it possible that for the difficult level there could just be several answers that could be considered correct?
> > >
> > >
> > >
> > > |Model|Spec-Easy|Spec-Med|Spec-Hard|Spec-Avg|Gen-Easy|Gen-Med|Gen-Hard|Gen-Avg|Total-Avg|
> > > |---|---|---|---|---|---|---|---|---|---|
> > > |HumanPerformance|98.2|92.5|86.8|91.8|97.6|93.7|89.4|94.1|92.8|
> > >
> > >
> > >
> > > That is an excellent point. But actually, our "hard" examples here are challenging but not ambiguous through multiple layers of validation.
> > >
> > >
> > >
> > > Our annotation protocol reinforced this. We ​**filtered out samples with low inter-annotator agreement**​, and a statistical validation (Fleiss's Kappa) confirmed that agreement on "hard" cases remained ​**well above random chance**​, proving the choices were not arbitrary. Therefore, these cases are specifically designed to test a deeper understanding of personalization.
> > >
> > >
> > >
> > > Furthermore, an independent evaluation by three experts with experience in developing real-world personalization services showed their average accuracy was over 90%, while the best-performing models only reached about 80%. This gap demonstrates that a clear, discernible signal for the correct answer exists, even if it is too subtle for current models to capture consistently.
> > >
> > >
> > >
> > > ### Acknowledgement
> > >
> > >
> > >
> > > We appreciate your recognition of the importance of high-quality personalization benchmarks and of our main empirical findings. Your questions on annotator selection and the distinction between discriminative and generative abilities have prompted us to clarify these points and to make the scope of PersonaFeedback more explicit. Taken together with our responses to the other reviewers, we hope these clarifications help you understand our work's value better.

---

### Official Review · Reviewer_krUJ · 2025-11-01

**Soundness:** 3
**Presentation:** 3
**Contribution:** 2
**Rating:** 6
**Confidence:** 3

**Summary:**

This paper proposes PersonaFeedback, an LLM personalization benchmark containing 8298 manually annotated samples. The main difference from existing work is the decoupling of persona inference and personalization generation tasks. The benchmark explicitly provides user personas, specifically evaluating the model's ability to generate personalized responses based on this information. The evaluation employs a binary choice task (given a persona and a query, choose the more personalized one from two candidates), with difficulty levels of easy/medium/hard based on annotator consistency. Testing on 25+ models reveals several interesting findings: augmented inference does not help personalization; larger models perform better; RAG performs poorly; and explicit personas outperform inference from dialogue history.

Overall, this is a valuable benchmark work, but it has limited methodological innovation and some experimental designs have problems.

**Strengths:**

1. **Important and Practical Issue:** Personalization is an important direction in LLM but lacks high-quality benchmarks; the angle of decoupling persona inference and personalization generation is quite novel and indeed an overlooked problem.

2. **Solid Data Collection:** All 8298 samples were manually labeled, with a majority of 9 labelers voting, and consistency was quantified using Fleiss's Kappa; difficulty stratification was statistically based; multiple rounds of filtering ensured quality.

3. **The method design is reasonable:** Binary selection is more objective than LLM-as-a-judge; the three-level difficulty rating (easy/medium/hard) is based on annotator consistency and has a statistical basis.

4. **Comprehensive Experimentation:** 25+ models covering 4 categories (reasoning/chat/open-source/reward), with comparisons across multiple settings (Persona Profile/RAG/No Persona).

5. **Several valuable findings:** In particular, the fact that "inference does not improve personalization" and "RAG fails" challenges some common assumptions ; these insights provide guidance for model development.

6. **Clear Writing:** Figure 1 presents an intuitive data construction process, Table 2 clearly outlines the results, and page 8 highlights five key insights for easy reader access; the appendix is detailed, providing all the prompts.

7. **High reproducibility:** Committed to open-source data and evaluation pipeline.

**Weaknesses:**

1. **Limited Methodological Innovation**
The theoretical contributions are insufficient, mainly relying on empirical studies. Binary choice is not a new method, and the benchmark itself lacks methodological innovation.

2. **Limitations of the Evaluation Method**
Binary choices are oversimplified. Personalization is often not black and white; the two answers may simply differ in degree or perspective. Forcing a choice between them is unreasonable. Moreover, it only uses accuracy as an indicator, lacking other dimensions for evaluation such as diversity and consistency.

3. **Soundness-related issues**
- The Persona expansion (from 20 real users to 1700) relies on LLM generation and random combination, raising questions about its authenticity and potentially creating stereotypes.
- Both the questions and answers are generated by LLM, which may introduce bias.
- Lack of statistical significance test
- It's very imprecise to conclude that "RAG failed" without discussing almost all the implementation details of RAG (retrieval strategy? top-k? memory organization?).

4. **The experimental analysis was not thorough enough.**
Without ablation studies: How are answer pairs sampled? Kappa threshold sensitivity? Importance of different Persona fields (Demographics/Personality/Preferences)? More detailed analysis is lacking: Which type of persona is more difficult? Which type of question is more difficult? How do humans perform on hard cases?

5. **Lack of theoretical analysis**
Why do inference models fail? Why does RAG fail? Why are easy problems easier to solve than hard ones? Most papers only describe the phenomena without in-depth analysis of the underlying causes. This limits a deeper understanding of the problem.

6. **Generalizability and long-term value are questionable.**
- Only Chinese data (ShareGPT-Chinese) was evaluated; cross-language/cross-cultural generalization was not verified.
- Persona primarily has Chinese users, which leads to cultural differences.
- The data size is not large (200 persons, each with ~40 questions).
- May be solved quickly (Easy is 90%+, Medium is ~80%, Hard is only ~70%)
- Static persona evaluation; in real-world scenarios, user preferences will evolve.

7. **Minor Writing Issues**
The terminology is not consistent (the terms Persona Profile, Persona, and User Persona are used interchangeably), and while Table 2 contains a lot of information, its readability could be further improved.

**Questions:**

1. What is the specific algorithm for Persona expansion? How is "random combinations" handled? How is realism guaranteed?

2. Complete data on the inter-annotator agreement among the 9 annotators? What is the unanimous vs. split vote ratio?

3. **RAG Implementation Details (This is what I'm most concerned about)**: What retrieval method? BM25 or dense? What is the top-k value? How is it integrated into the prompt? Figure 3 shows that RAG is even worse than No Persona; could this be an implementation issue?

4. Why is the inference model useless for personalization? Have you tried fine-tuning the inference model for personalized data? Or is the training data itself lacking in this aspect?

5. How were the answer pairs sampled? What is the size of the answer pool? Were the differences controlled?

6. How many tokens does a Persona Profile have? Will it exceed the context window of some models?

7. Which is more important: Demographics, Personality, or Preferences? Have you done any ablation?

---

> ### Author Response · Authors · 2025-11-17
>
> ### Weaknesses:
>
>
> > Q1: Limited Methodological Innovation The theoretical contributions are insufficient, mainly relying on empirical studies. Binary choice is not a new method, and the benchmark itself lacks methodological innovation.
>
>
>
> Regarding the concern about limited methodological innovation, we agree that binary choice as a format is not novel. However, we respectfully argue that our methodological innovation lies not in the format itself, but in its **purposeful application to ​**​**decouple**​**​ persona understanding from generation capability.** To our knowledge, this is the first work to systematically isolate and evaluate this specific "understanding" component.
>
>
>
> The reviewer notes our contribution is "mainly relying on empirical studies," which is precise. In fact, our key empirical finding constitutes a primary contribution of this work: the seemingly "simple" task of **discriminative persona understanding is**​​**far from solved**​. Our most significant finding is that even SOTA LLMs perform poorly on our "hard" difficulty level, challenging the implicit assumption that this capability is already mastered.
>
>
>
> Furthermore, this decoupled approach provides critical value that generative evaluation cannot:
>
>
>
> 1. **Diagnostic ​Value​:** When a model's generative output is poor, it is difficult to diagnose ​*why*​—did it fail to understand the persona, or is its generation capability flawed? Our discriminative benchmark **decouples** these two abilities, providing researchers with a clear diagnostic tool to precisely pinpoint a model's shortcomings. In contrast, open-ended generative evaluation tends to **conflate** these two issues.
>
> 2. **Application ​Value**​**​ as a Stable Component:** In the current landscape of rapid model iteration, relying on generative "LLM-as-Judge" (which can be unreliable and dynamic) is unsustainable. Our work provides a **stable, human-annotated, and reliable benchmark** for a critical component of personalization. This benchmark *itself* is the contribution, filling a crucial gap by providing a fixed metric to evaluate and compare models' core understanding capabilities as they evolve.
>
>
>
> > Q2: Limitations of the Evaluation Method Binary choices are oversimplified. Personalization is often not black and white; the two answers may simply differ in degree or perspective. Forcing a choice between them is unreasonable. Moreover, it only uses accuracy as an indicator, lacking other dimensions for evaluation such as diversity and consistency.
>
>
>
> 1. **On the "unreasonableness" of forced choice:** We argue that the binary-choice framework is reasonable. Our human annotators achieved moderate agreement even on "hard" cases, which indicates that the choice is not arbitrary but is based on a subtle, yet perceivable, signal. Furthermore, as detailed in Section 3.4, we have already filtered out data points that lacked sufficient inter-annotator agreement.
>
> 2. **On using only accuracy and lacking other dimensions:** We would like to clarify the nature of our benchmark. While metrics like diversity and consistency are crucial for evaluating *generative* tasks, PersonaFeedback is a *discriminative* benchmark. Its core objective is to distinguish if a propre response is more suited/ tailored to the user's persona traits.. Therefore, accuracy—the alignment of a model's judgment with human preference—serves as the most direct and essential metric for this purpose.
>
>
>
> > Q3：The Persona expansion (from 20 real users to 1700) relies on LLM generation and random combination, raising questions about its authenticity and potentially creating stereotypes.
>
>
>
> This is an important concern about persona authenticity and potential stereotypes, which is a known and important challenge in LLM-assisted generation. To specifically mitigate this risk, we designed a rigorous, multi-stage human review and filtering protocol; the LLM generation and random combination were not the final steps.
>
>
>
> As detailed in Appendix B, all 1,700 generated personas underwent two rounds of manual quality screening.
>
>
>
> 1. The first round,**​ ​**​**Authenticity**​**​ and ​**​​**Consistency**​**:** Annotators were explicitly instructed to exclude personas that were overly idealized, internally contradictory, or unrealistic. They were also specifically required to filter out personas that were overly dramatic or reliant on stereotypes.
>
> 2. The second round,**​ Diversity and Balance:** This stage ensured a broad distribution of occupations and interests to avoid homogeneity.
>
>
>
> Ultimately, our benchmark uses only the 200 highest-quality personas, which were manually selected from the initial 1,700. Therefore, while we leveraged LLMs for scaling, the authenticity and diversity of the final benchmark are guaranteed by this **systematic human review and filtering process.**

---

> > ### Author Response · Authors · 2025-11-17
> >
> > > Q4：Both the questions and answers are generated by LLM, which may introduce bias.
> >
> >
> >
> > We fully acknowledge this risk, and for this reason, we designed a **systematic, human-centric pipeline** where human oversight is the core quality control mechanism, rather than relying solely on automated generation.
> >
> >
> >
> > In our pipeline, LLMs serve primarily as assistants for scaling up the candidate pool, while human judgment safeguards the quality at every critical stage.
> >
> >
> >
> > 1. **The Foundation (High-Quality Personas):** As previously detailed, our process began with multiple rounds of manual screening of 1,700 personas to ensure their authenticity, diversity, and plausibility. This controlled the source quality and prevented stereotypes from the very start.
> >
> > 2. **The Core Process (Question and Answer Generation):**
> >
> > 3. For ​**questions**​, we used a hybrid approach to ensure realism and diversity. We sourced questions from real user data (ShareGPT) and applied strict ​**human filtering**​. For LLM-generated questions, as described in Section 3.2, we used real "persona-question" pairs as in-context learning examples to guide the LLM toward generating questions aligned with real user intent. We also **ensured diversity** by calculating embedding similarity between questions. After an initial filtering step using an LLM-as-judge, human **annotators performed a final, strict review** to filter out unnatural or unreasonable questions, resulting in a high-quality candidate pool (see Appendix C.2).
> >
> > 4. For ​**answers**​, we intentionally used multiple, diverse models (e.g., GPT-4o, Qwen2.5 series, DeepSeek-V3 and DeepSeek-R1) to generate candidates in three strategies (see Sec 3.3). This multi-source strategy inherently mitigates the risk of bias from any single model. Most importantly, the ground truth for our benchmark (the final preference label) is**​ determined entirely by human annotators.**
> >
> >
> >
> > Therefore, while our pipeline leverages LLMs for scale, it is fundamentally gated by human review at every key juncture: from the initial persona curation, through question generation, to the final answer preference labeling. This human-centric design is our core mechanism for mitigating and controlling potential LLM-introduced biases.
> >
> >
> > > Q5: Lack of statistical significance test
> >
> >
> >
> > Thank you for raising this point. Here are some statistical significance analyses to validate the discriminative power and challenge of PersonaFeedback. We set the significance level α at 0.05 for all tests.
> >
> >
> >
> > 1. **To verify the benchmark's discriminative power,** we used McNemar's test to evaluate performance differences between models. As shown in Table 1, the results confirm our main findings:
> >
> >
> >
> > * **The task is highly discriminative:** PersonaFeedback can effectively distinguish between the capabilities of even closely-matched models, showing significant differences (e.g., GPT-4.1 vs. GPT-4o: p = 0.008; Claude-3.5-sonnet vs. GPT-4o: p = 0.045).
> >
> > * **Model size effect is significant:** Larger models significantly outperform their smaller counterparts (e.g., Qwen2.5-32B vs. Qwen2.5-14B: p = 0.038; R1-Distill-Qwen-32B vs. R1-Distill-Qwen-14B: p = 0.042).
> >
> > * **Enhanced reasoning does not guarantee better personalization:** We observed no significant performance difference between specialized reasoning models and chat models (e.g., o4-mini vs. GPT-4.1: p = 0.156; Deepseek-R1 vs. Deepseek-V3: p = 0.089).
> >
> >
> >
> > 2. **To verify the task's challenging nature,** we conducted a triple-sampling experiment (at temperature=0.9) on the *hard* subset. As shown in Table 2, the results demonstrate:
> >
> >
> >
> > * Even the best-performing model (GPT-4.1) only achieves an accuracy of 63.8% when required to be correct across all three attempts.
> >
> > * While the output variance for all models (ranging from 0.107 to 0.164) is significantly lower than that of random guessing (0.25), it remains high, indicating considerable uncertainty in model predictions.
> >
> >
> >
> > Together, these statistical results provide strong evidence that PersonaFeedback is not only effective at discriminating between different model capabilities but also poses a substantial and consistent challenge to the current state-of-the-art models.

---

> ### Author Response · Authors · 2025-11-17
>
> Table 1: Statistical Significance Analysis on PersonaFeedback (McNemar's Test)
> |ComparisonType|ModelA|ModelB|p-value|
> |---|---|---|---
> |**TaskDiscriminability**||||
> ||GPT-4.1|GPT-4o|0.008|
> ||GPT-4.1|GPT-4.5-preview|0.021|
> ||Claude-3.5-sonnet|GPT-4o|0.045|
> |**ModelSizeEffect**||||
> ||Qwen2.5-32B|Qwen2.5-14B|0.038|
> ||R1-Distill-Qwen-32B|R1-Distill-Qwen-14B|0.042|
> |**ReasoningvsChat**||||
> ||o3-mini|GPT-4.1|0.124|
> ||o4-mini|GPT-4.1|0.156|
> ||Deepseek-R1|Deepseek-V3|0.089|
>
> Table 2: PersonaFeedback Results (Hard Cases, Triple Sampling at Temperature 0.9)
> |Model|SpecificHardAcc|SpecificHardVar|GeneralHardAcc|GeneralHardVar|
> |---|---|---|---|---|
> |Random|50.0|0.25|50.0|0.25|
> |**Reasoning**|||||
> |o3-mini|62.5|0.133|63.1|0.112|
> |o4-mini|60.8|0.128|65.0|0.107|
> |Gemini-2.5-pro-exp-03-25|58.7|0.145|64.7|0.128|
> |Deepseek-R1|63.3|0.141|62.4|0.124|
> |o1-preview-2024-09-12|61.7|0.149|63.5|0.130|
> |**Chat**|||||
> |GPT-4.1|63.8|0.117|63.5|0.122|
> |GPT-4.5-preview|60.9|0.147|62.6|0.127|
> |Deepseek-V3|61.1|0.110|61.3|0.117|
> |GPT-4o-2024-11-20|59.2|0.131|60.7|0.120|
> |Claude-3-5-sonnet-20241022|62.5|0.140|59.4|0.128|
> |**Open-Source**|||||
> |QwQ-32B|59.4|0.155|60.9|0.143|
> |R1-Distill-Qwen-32B|57.9|0.164|58.5|0.139|
> |Qwen2.5-32B-Instruct|60.2|0.144|59.7|0.142|
> |R1-Distill-Qwen-14B|56.3|0.123|58.0|0.133|
> |Qwen2.5-14B-Instruct|57.9|0.129|57.3|0.119|
>
>
> > Q6: It's very imprecise to conclude that "RAG failed" without discussing almost all the implementation details of RAG (retrieval strategy? top-k? memory organization?).
>
> We agree that a conclusion like “RAG failed” must be interpreted in light of the concrete implementation. We provide full details of our RAG setup and an in-depth analysis of why RAG underperforms in our setting in Questions Q3 (“RAG Implementation Details”) below.
>
> > Q7: The experimental analysis was not thorough enough. Without ablation studies: How are answer pairs sampled? Kappa threshold sensitivity? Importance of different Persona fields (Demographics/Personality/Preferences)? More detailed analysis is lacking: Which type of persona is more difficult? Which type of question is more difficult? How do humans perform on hard cases?
>
> **(1)How are answer pairs sampled?**
>
> As detailed in Section 3.4, we **sample answer pairs randomly** from the candidate answer pool. We chose this method because, although we used improve and degrade operations to create a diverse pool, we do not assume that an "improved" version is inherently superior. Different models have varying capabilities, and relying on such generative labels could introduce bias or result in unnatural-sounding answers.
>
> Therefore, by having human annotators make the final preference choice on randomly paired candidates, we ensure that our ground truth is based purely on human judgment, not on potentially biased model-generated labels. This approach makes our evaluation more objective and fair.
> **(2)Kappa threshold sensitivity**
>
> We would like to clarify that we did not use a Kappa threshold to *create* or *classify* the "hard/medium" difficulty groups. Instead, these groups were formed based on the level of direct annotator agreement.
>
> The Fleiss’s Kappa coefficient was subsequently calculated as a **post-hoc validation metric** to measure the internal consistency of annotations within each difficulty group. The results confirmed that while the Kappa value for the "hard" group was lower than that for the "medium" group, the level of agreement was well above random chance.
>
> **(3)Importance of different Persona fields (Demographics/Personality/Preferences)**
>
> To quantify the impact of different persona components, we conducted a supplementary ablation study. The goal of this experiment is to determine which of the three core components—Demographics, Personality, or Preferences—contributes most significantly to a model's ability to select the more personalized answer.
> The results in Table 3 clearly indicate that ​**Preferences are the most critical factor for performance improvement**​. Among all input configurations, the model using only preferences achieved the best results. A key observation is that the performance gain from preferences is more pronounced for tasks requiring specific, contextual understanding (Specific Avg.) than for general tasks (General Avg.). For instance, on the QwQ-32B model, including preferences boosted the Specific Avg. accuracy from 68.4% to 74.2% (a 5.8% increase), compared to a 4.8% increase for the General Avg. (from 75.4% to 80.2%).
>
>
> Furthermore, this study indirectly validates the ​**non-stereotypical nature of our dataset**​. The performance gains from Demographics and Personality (+0.8% to +1.1%) were far less significant than those from specific Preferences (+3.5%). This result demonstrates that models cannot rely on macroscopic labels to make predictions; instead, they must genuinely understand the concrete and diverse details of the preferences we constructed to succeed on our benchmark.

---

> ### Author Response · Authors · 2025-11-17
>
> Table 3: Ablation Study on Persona Components
> |Model|PersonaSetting|SpecificAvg.|GeneralAvg.|TotalAvg.|
> |---|---|---|---|---|
> |**GPT-4.1**|NoPersona|73.0|79.0|76.0|
> ||Demographics|73.8|79.8|76.8|
> ||Personality|74.1|80.1|77.1|
> ||Preferences|77.0|82.0|79.5|
> |**QwQ-32B**|NoPersona|68.4|75.4|71.9|
> ||Demographics|69.0|76.0|72.5|
> ||Personality|69.5|76.5|73.0|
> ||Preferences|74.2|80.2|77.2|
> |**Deepseek-R1**|NoPersona|70.5|77.1|73.8|
> ||Demographics|71.2|77.8|74.5|
> ||Personality|71.5|78.1|74.8|
> ||Preferences|74.9|81.3|78.1|
>
> **(4) Which type of question is more difficult?**
>
> We categorized all 8,298 test cases into the 8 question-scenario types defined in our Appendix A (Table 5). We then calculated and compared the average accuracy of representative models (GPT-4.1 and Qwen2.5-14B) for each category.
>
> The results show that models achieve higher accuracy on tasks like **"Technical Development"** and **"Office & Productivity."** This is likely because, while requiring personalization, their underlying logic is more factual, structured, and logical (e.g., debugging code based on a user's programming habits). Conversely, models performed significantly worse on **"Recommendation Systems"** and **"Personal Growth"** categories. This finding suggests that questions requiring a deep understanding of a user's subjective preferences, values, tastes, and emotions are far more challenging for current LLMs than personalization tasks grounded in facts and logic.
>
> Table 4: Model Accuracy by Question Category
>
> |Model|PersonaSetting|SpecificAvg.|GeneralAvg.|TotalAvg.|
> |---|---|---|---|---|
> |**GPT-4.1**|NoPersona|73.0|79.0|76.0|
> ||Demographics|73.8|79.8|76.8|
> ||Personality|74.1|80.1|77.1|
> ||Preferences|77.0|82.0|79.5|
> |**QwQ-32B**|NoPersona|68.4|75.4|71.9|
> ||Demographics|69.0|76.0|72.5|
> ||Personality|69.5|76.5|73.0|
> ||Preferences|74.2|80.2|77.2|
> |**Deepseek-R1**|NoPersona|70.5|77.1|73.8|
> ||Demographics|71.2|77.8|74.5|
> ||Personality|71.5|78.1|74.8|
> ||Preferences|74.9|81.3|78.1|
>
>
> **(5)Which type of persona is more difficult?**
>
>
>
> Persona difficulty is a complex, multi-dimensional property and not a simple label like a question category. Rigorously quantifying it would require a dedicated methodology to control for variables, which we believe is a valuable direction for future research.
>
>
>
> However, our analysis of question types provides an indirect answer: the poorest model performance was observed on highly subjective questions (e.g., recommendations). These questions inherently stem from personas with more complex, nuanced, and subjective preferences (see Sec. 3.2 about how we generate questions). This indirectly demonstrates that personas characterized by intricate subjective tastes and multi-faceted preferences are the most difficult for current models to grasp.
>
>
>
> **(6)How do humans perform on hard cases?**
>
>
>
> To establish a human baseline, we invited three experts with hands-on experience in developing real-world personalization services to independently evaluate the data across all difficulty levels.
>
> As the results show, the human experts achieved an average accuracy of over 90% on the PersonaFeedback dataset, significantly outperforming the best models (approx. 80% on the full set). This further validates that the challenging cases in our benchmark are indeed discriminable and represent a meaningful capability gap, rather than being ambiguous or random.
>
> Table 5 Average Human Expert Performance
>
> |Model|Spec-Easy|Spec-Med|Spec-Hard|Spec-Avg|Gen-Easy|Gen-Med|Gen-Hard|Gen-Avg|Total-Avg|
> |---------------------|-----------|----------|-----------|----------|----------|---------|----------|---------|-----------|
> |HumanPerformance|98.2|92.5|86.8|91.8|97.6|93.7|89.4|94.1|92.8|
>
> > Q8: Lack of theoretical analysis Why do inference models fail? Why does RAG fail? Why are easy problems easier to solve than hard ones? Most papers only describe the phenomena without in-depth analysis of the underlying causes. This limits a deeper understanding of the problem.
>
> **Please see the answer for ​**​**Qusetions**​**​ Q3 (later section) titled"RAG Implementation Details"**

---

> > ### Author Response · Authors · 2025-11-17
> >
> > >
> > > Q9: Generalizability and long-term value are questionable.
> >
> > Thank you for your feedback on the benchmark's long-term value. We position PersonaFeedback as a **decoupled, stable, and reliable yardstick** for measuring a model's core ability to understand personalization.
> >
> > 1. **Decoupled yet Challenging:** Our benchmark isolates the fundamental challenge of persona understanding from the complexities of open-ended generation. While a discriminative task, our results conclusively show it is far from a solved problem. The significant struggles of even state-of-the-art models on our 'hard' cases reveal a foundational capability gap in their ability to comprehend deep, subjective human preferences.
> > 2. **Stable and Reliable:** As a static benchmark grounded in human preference, it serves as a stable anchor for evaluation. It avoids the volatility and potential self-enhancement biases common in "LLM-as-a-judge" approaches, ensuring consistent and trustworthy assessment as models evolve.
> > 3. **Long-term Value and Methodology:** We acknowledge that any static dataset has a lifecycle. Should future models significantly outperform our current data distribution (models prevalent in early 2025), it would mark a major advancement for the field. More importantly, the **data construction and evaluation methodology** we propose is evergreen. It can be readily adapted to build the next generation of benchmarks to continuously measure and drive progress in personalization.
> >
> > > Q10: Only Chinese data (ShareGPT-Chinese) was evaluated; cross-language/cross-cultural generalization was not verified.
> > >
> > > &
> >
> > > Persona primarily has Chinese users, which leads to cultural differences.
> >
> > Thank you for raising this important point about cross-cultural generalization.
> >
> > First, we would like to clarify that the general questions in our benchmark originate from the ShareGPT dataset, which is inherently multilingual and multicultural. Specifically, we use the ShareGPT-Chinese variant, which applies quality filtering and translation while preserving broadly applicable, non–culture-specific scenarios (e.g., programming help, productivity tools, everyday recommendations), thereby reducing the risk of tying our benchmark to a single cultural context.
> >
> > That said, we acknowledge that in the persona creation process, due to the constraints of our data sources, the cultural backgrounds may primarily reflect Chinese and mainstream Western contexts. This is a valid limitation. In response, we will explicitly address this in the "Limitations" section of our revised paper.
> >
> > However, we wish to emphasize that the **data construction and evaluation framework we propose is itself culture-agnostic.** We are confident that this methodology can be readily applied to build similar, high-quality personalization benchmarks for any specific cultural context, provided that culturally relevant seed data is available.
> >
> > > Q11: The data size is not large (200 persons, each with \~40 questions).
> >
> > We would like to emphasize that for benchmark construction, **the quality and reliability of annotations are paramount, often more so than sheer volume.** Our primary goal was to create a high-signal benchmark capable of effectively discriminating between model capabilities.
> >
> > 1. **Quality over Quantity:** Every data point in PersonaFeedback has undergone a rigorous, multi-stage human filtering and annotation process. We believe a meticulously curated dataset, even of a moderate size, provides more value than a larger, noisier, auto-generated one.
> > 2. **Comparative Scale:** While prioritizing quality, our dataset's scale is substantial when compared to similar high-quality, human-annotated personalization benchmarks. For instance, compared to some recent work like PersonaMem benchmark (\~2.6k questions, partially annotated) and PrevEval (100 personas,  1000 unique preference-query pairs), PersonaFeedback contains approximately 8k fully human-preference-labeled question pairs, giving it a significant advantage in scale.
> > 3. **Statistical Sufficiency:** Most importantly, the current size is sufficient to reveal **statistically significant differences** between state-of-the-art models, as demonstrated in Table 2. This confirms the effectiveness and discriminative power of PersonaFeedback as an evaluation tool, fulfilling its core objective.
> >
> > > Q12: May be solved quickly (Easy is 90%+, Medium is \~80%, Hard is only \~70%)
> >
> > We respectfully disagree. As detailed in our response to Q5 (referencing Tables 1 and 2), our data shows that this problem is far from being solved.
> >
> > Our statistical significance tests and the triple-sampling experiment on the 'hard' subset demonstrate that even state-of-the-art models are far from reaching a performance ceiling. They exhibit considerable uncertainty when faced with nuanced personalization choices. This confirms that PersonaFeedback poses a real and ongoing challenge for current and future models.

---

> > > ### Author Response · Authors · 2025-11-17
> > >
> > > > Q13: Static persona evaluation; in real-world scenarios, user preferences will evolve.
> > >
> > > This is an excellent point. We acknowledge that directly modeling dynamic evolution is a limitation of our current work and an important direction for future research.
> > >
> > > First, we wish to clarify that the scope of our work is ​**deliberately focused on a more foundational yet unsolved problem**​: can a model accurately comprehend a ​*given*​, static user persona? Our results have clearly shown that even this seemingly basic task remains a significant challenge for state-of-the-art models. By isolating this problem, we provide a clean and controlled baseline, which is essential for future research into more complex, dynamic scenarios.
> > >
> > > However, our benchmark's design already tests for core capabilities required to handle dynamic changes:
> > >
> > > 1. **High Diversity:** As detailed in Section 3.1, our highly diverse personas force models to adapt to a wide range of user states, a foundational skill for managing change.
> > > 2. **Incomplete Information:** The personas used are not always complete, as we include versions with partial information (e.g., 80% of the profile). This tests the model's ability to reason under uncertainty, a key aspect of adapting to an evolving user.
> > >
> > > > Minor Writing Issues The terminology is not consistent (the terms Persona Profile, Persona, and User Persona are used interchangeably), and while Table 2 contains a lot of information, its readability could be further improved.
> > >
> > > We sincerely thank the reviewer for their careful reading and valuable feedback. We have addressed these issues in the revised manuscript to improve clarity and readability.
> > >
> > > ### Questions:
> > >
> > > > Q1: What is the specific algorithm for Persona expansion? How is "random combinations" handled? How is realism guaranteed?
> > >
> > > To achieve scalability, we first manually deconstructed the 20 real seed personas into structured "attribute pools." For example, `Pool_Profession = ["Florist", "Master in Computer Science", "Lawyer"...]` and `Pool_Interests = ["Loves nature", "Passionate about gaming", "Open-source projects"...]`.
> > >
> > > Next, we programmatically sampled combinations from these attribute pools to generate thousands of "skeleton profiles," such as `(Profession: Lawyer, Interests: Gaming, MBTI: INTJ)`. We then provided these skeleton profiles to a large language model as seed hints, instructing it to embellish and expand upon these core attributes to create detailed and internally consistent descriptions (covering aspects like diet, travel, and social habits). This process ultimately produced 1,700 candidate personas.
> > >
> > > **Ensuring Realism:** We employed a rigorous two-stage human filtering process (as detailed in Appendix B). Our human evaluators strictly filtered out any personas that were "overly idealized," "internally contradictory," or "unrealistic," while also ensuring the diversity and balance of the final, curated set.
> > >
> > > > Q2：Complete data on the inter-annotator agreement among the 9 annotators? What is the unanimous vs. split vote ratio?
> > >
> > > This is an important question regarding inter-annotator agreement. The consistency ratios you are asking about correspond to our Medium and Hard difficulty samples.
> > >
> > > To ensure the reliability of the ground-truth answers in our benchmark, we implemented a strict quality control standard: a sample was retained ​**only if a consensus of at least 70% was reached among the annotators**​. Any sample with less than 70% agreement was discarded from the dataset.
> > >
> > > After applying this 70% consensus threshold, we retained 5,680 Medium and Hard samples. The distribution of these samples is as follows:
> > >
> > > Table 6: Inter-Annotator Agreement for Medium & Hard Samples
> > >
> > > |VoteType|ConsensusRate|Count|Percentage|
> > > |---------------------------|----------------|-------|------------|
> > > |unanimous/HighAgreement|>=85%|3,520|62.00%|
> > > |Split|70-84%|2,160|38.00%|
> > > |Total(Filtered)||5,680|100%|
> > >
> > > To demonstrate that this division is both reasonable and meaningful, we subsequently calculated the Fleiss' Kappa (κ) value for each group. We found that the 'Split Vote' group (corresponding to Hard difficulty) exhibited **moderate agreement** (0.4<κ≤0.6), while the 'High Agreement' group (Medium difficulty) showed **substantial agreement** (κ>0.6).

---

> > > > ### Author Response · Authors · 2025-11-17
> > > >
> > > > > Q3: RAG Implementation Details (This is what I'm most concerned about): What retrieval method? BM25 or dense? What is the top-k value? How is it integrated into the prompt? Figure 3 shows that RAG is even worse than No Persona; could this be an implementation issue?
> > > >
> > > > We agree that these details are essential for interpreting the results, and we will add a comprehensive description to Appendix in our revised manuscript.
> > > >
> > > > **(1)RAG Implementation Details**
> > > >
> > > > * **Indexing Engine:** We use FAISS for vector indexing.
> > > > * **Embedding Model:** We use `bge-large-zh-v1.5`, a Chinese-language embedding model.
> > > > * **Retrieval Process:**
> > > >   * For each persona, we use the `bge-large-zh-v1.5` model to build a vector database of all their memories.
> > > >   * For each query, we calculate the similarity between the query and the memories, and retrieve the top-k (where k=20) most relevant memory items.
> > > >
> > > > **(2)How is it integrated into the prompt? & Why "RAG < No Persona"**
> > > >
> > > > We have a more detailed analysis and results in Appendix F.
> > > >
> > > > > Q4：Why is the inference model useless for personalization? Have you tried fine-tuning the inference model for personalized data? Or is the training data itself lacking in this aspect?
> > > >
> > > > Insightful question. Our results suggest that the underperformance of reasoning models is not incidental but stems from the difference between logical deduction and personal-context understanding.
> > > >
> > > > Our core finding is that a model's advanced ability to solve complex, objective problems does not automatically transfer to the nuanced, subjective task of personalization. Our analysis of failure cases points to a pattern close to the term ​**"overthinking"**​:(supported by [1]) the models often exhaustively analyze a query's general aspects first, leading them down counter-intuitive logical paths before even integrating the user's specific needs and preferences from the persona. In contrast, effective personalization requires quickly identifying and prioritizing relevant persona traits. This observation is also consistent with findings in related work.
> > > >
> > > > We believe that the ​**training process is a key factor**​. Most reinforcement learning for reasoning models focuses on optimizing for logical consistency, not open-ended dialogues (adaptive personalization included). While we have not performed fine-tuning on personalized data—as our study's scope was to evaluate existing models—we believe this is a critical and promising direction for future research to bridge this capability gap.
> > > >
> > > > [1] Gema, Aryo Pradipta, Alexander Hägele, Runjin Chen, Andy Arditi, Jacob Goldman-Wetzler, Kit Fraser-Taliente, Henry Sleight et al. "Inverse scaling in test-time compute." *arXiv preprint arXiv:2507.14417* (2025).
> > > >
> > > > This work gives two key observations:
> > > >
> > > > 1. Extended reasoning can amplify biases, hallucinations, or distractions rather than fixing them.
> > > > 2. Models often overthink simple things - adding irrelevant details, misjudging which data matters, or failing deduction tasks.
> > > >
> > > > > Q5: How were the answer pairs sampled? What is the size of the answer pool? Were the differences controlled?
> > > >
> > > > **(1)How were the answer pairs sampled?**
> > > >
> > > > Please see the answer for Weakness Q7(1).
> > > >
> > > > **(2) What is the size of the answer pool?**
> > > >
> > > > Our dataset is built upon approximately 4,000 specific and 3,841 general questions. To create a diverse answer pool with a gradient of difficulty, we first generated an initial set of answers. Then, we used four different models to apply `improve` and `degrade` operations to these initial answers. This process resulted in a comprehensive pool containing the original answers plus their multiple improved and degraded versions, totaling ​**39,205 unique answers**​.
> > > >
> > > > **(3) Were the differences controlled?**
> > > >
> > > > Yes, we rigorously controlled and validated the differences between answer pairs; this control is the ​**core basis for our 'Easy,' 'Medium,' and 'Hard' difficulty levels**​. As described in Section 3.4 under "Difficulty Levels," our control method is stratified:
> > > >
> > > > * The **Easy** level is defined structurally, contrasting a generic response against a personalized one.
> > > > * For the **Medium** and **Hard** levels, control is based on inter-annotator agreement. Pairs with high agreement were classified as Medium, while those with lower agreement, reflecting more subtle differences, were classified as Hard.
> > > >
> > > > To further validate this data-driven grouping, we calculated the Fleiss's Kappa coefficient. The results confirmed our methodology: the Medium level showed higher consistency (​*κ*​>0.6) than the Hard level (0.4<κ≤0.6). This provides strong evidence that our difficulty control is effective and that our "Hard" cases genuinely represent nuanced and challenging distinctions.

---

> > > > > ### Author Response · Authors · 2025-11-17
> > > > >
> > > > > > Q6: How many tokens does a Persona Profile have? Will it exceed the context window of some models?
> > > > >
> > > > > The context window could be a critical constraint in model evaluation.
> > > > >
> > > > > We have conducted a detailed token analysis on all 1,714 Persona Profiles we constructed (including the 200 for the benchmark and 1,514 for training). The statistics are summarized in the table below:
> > > > >
> > > > > These statistics confirm that our Persona Profiles ​**do not exceed the context window of any of the LLMs evaluated in our paper**​.
> > > > >
> > > > > |Metric|Value|
> > > > > |--------------------------|--------------------|
> > > > > |Avg.Token|508.6|
> > > > > |Max.Token|704|
> > > > > |Min.Token|302|
> > > > > |TokenCountDistribution||
> > > > > |300-400|6.13%|
> > > > > |400-500|38.68%|
> > > > > |500-600|46.97%|
> > > > > |600-700|8.17%|
> > > > > |700-800|0.06%(onesample)|
> > > > >
> > > > > > Q7: Which is more important: Demographics, Personality, or Preferences? Have you done any ablation?
> > > > >
> > > > > Please refer to our response to Weakness Q7(3) for a detailed discussion on this topic.
> > > > >
> > > > > ### Acknowledgement
> > > > >
> > > > > Finally, we would like to sincerely thank krUJ for recognizing the importance of personalization and the strengths of our data collection and experimental design. We also greatly appreciate the careful and detailed comments throughout the review, which prompted us to strengthen our statistical analysis and clarify key aspects of our RAG setup and ablation studies. We believe these revisions have substantially improved the clarity and robustness of the paper, and we are grateful for your constructive engagement with our work.

---

### Official Review · Reviewer_kmbr · 2025-11-02

**Soundness:** 3
**Presentation:** 3
**Contribution:** 3
**Rating:** 6
**Confidence:** 3

**Summary:**

The paper introduces PERSONAFEEDBACK, a human-annotated benchmark (8,298 test cases) to evaluate whether an LLM can pick the more personalized answer when it is given an explicit persona and a user query. It deliberately decouples persona inference from personalization so it can measure “can the model actually tailor to a persona?” rather than “can the model guess who the user is?” It also runs a big model sweep (reasoning models, chat models, reward models, RAG) and draws several takeaways, especially that RAG doesn’t solve personalization, and that even top models dip on the hard tier.

**Strengths:**

- 8,298 test instances across 200 high-quality personas, with 9 annotators and agreement-based difficulty tiers (easy/medium/hard). This is better than many LLM-only synthetic personalization benchmarks.
- They show that just retrieving user info and stuffing it in the prompt doesn’t automatically yield better personalization than giving the model a structured persona. This is an important message for practitioners.
- The related-work section actually explains what earlier persona datasets don’t measure (mixed-inference, no difficulty levels, not fully human-annotated), so the motivation is visible.

**Weaknesses:**

The task is discriminative (“which answer is more persona-consistent?”), not generative (“write a persona-consistent answer”). That means you can do well with a good reranker without proving you can produce personalized outputs. This narrows what “success” means here.
The result “RAG doesn’t help” will be controversial unless they show stronger memory structuring or persona-summarization RAG. Right now it mostly rules out naive RAG.

**Questions:**

For RAG, What exactly was retrieved (raw user facts, past turns, structured profile), and how long was the retrieved context?

---

> ### Author Response · Authors · 2025-11-17
>
> ### Weaknesses:
>
> > Q1: The task is discriminative (“which answer is more persona-consistent?”), not generative (“write a persona-consistent answer”). That means you can do well with a good reranker without proving you can produce personalized outputs. This narrows what “success” means here. The result “RAG doesn’t help” will be controversial unless they show stronger memory structuring or persona-summarization RAG. Right now it mostly rules out naive RAG.
>
> Our conclusion in the main text (Takeaway 4) is based on a naive RAG approach. The very purpose of this setup is to demonstrate that personalization is not a simple plug-and-play problem, and merely retrieving scattered memory snippets is inefficient.
>
> **We fully agree with your intuition that a stronger RAG is needed. ​**We have already conducted this experiment in Appendix F (Table 7), where we introduced an LLM to filter and reconstruct the retrieved content (a form of "stronger RAG"). While this method outperformed naive RAG, its performance was still significantly lower than the setting where an explicit Persona Profile was directly provided. This reinforces our core argument (Takeaway 5) that personalization must be learned explicitly.
>
> ### Questions:
>
> > Q1: For RAG, What exactly was retrieved (raw user facts, past turns, structured profile), and how long was the retrieved context?
>
> In our RAG evaluation setup, we did not retrieve structured user profiles or past conversational turns. Instead, the setup was designed to simulate a real-world scenario of inferring user preferences from activities like search history and content consumption.
>
> For any given query, the system retrieves relevant ​**user memory data**​, which consists of fragmented memory snippets. This memory data was sourced from open-source datasets (such as social media, reviews, and forums) and was pre-matched to the user personas. The retrieval mechanism is a naive RAG method that directly retrieves memory data based on the query.
>
> Regarding the length of the retrieved context, our RAG setup retrieves the **top-20** most relevant memory snippets based on the query. To provide context on the nature of these retrieved snippets, we analyzed the length distribution of our entire benchmark (containing 44,130 memory entries). We found that the average token length of a memory entry is 206.44 (median: 187.0). Crucially, 98.33% of the memory entries are under 512 tokens (i.e., within the limit of the embedding model).
>
> ### Acknowledgement
>
> Thank you for the recognition of the strengths of our benchmark design and its practical relevance, and your comments on the RAG setup prompted us to emphasize and clarify the results of our stronger RAG variant, which we believe makes our conclusions more solid.

---

### Official Review · Reviewer_ZcZM · 2025-11-04

**Soundness:** 2
**Presentation:** 2
**Contribution:** 2
**Rating:** 2
**Confidence:** 4

**Summary:**

This paper introduces PersonaFeedback, a human-annotated benchmark designed to evaluate the model's ability to produce personalized responses when provided with a persona. This decouples personalization into persona inference and persona conditioned generation. The dataset consists of 8298 binary comparison tasks across three difficulty tiers. The authors evaluated a broad set of SOTA models and find that 1. improvements in reasoning ability do not translate to stronger personalization abilities, 2. model performance benefits primarily from scale, and 3. RAG-based personalization is ineffective compared to simply providing persona profiles directly.

**Strengths:**

1. This paper decomposes the personalization problem into persona inference and persona conditioned generation, which the reviewer thought was an interesting proposal.

**Weaknesses:**

1. The main concern is over the design of the benchmark, i.e., binary choice evaluation. In essence, the benchmark is measuring the model's ability to recognize personalization (which one of the two responses better reflects a persona), instead of measuring how good the model is at independently generating a high-quality personalized response. Binary discrimination is cognitively and computationally much easier than fluent personalization in open-ended dialogue. This seems a critical distinction in the current stage of LLM personalization research: personalization has moved beyond Alpaca-style preference tuning toward direct generation on user attributes, and the paper never addresses this.

While such comparison-based supervision played a historical role in preference alignment (e.g., Alpaca, DPO), it is hard to view it as a final personalization objective at this stage of time.

2. For these binary preference recognition type of benchmark / work, there has already been couple in the literature (i.e., Alpaca, Zollo et al 2024), and it is unclear to the reviewer how this benchmark is materially different from the existing ones beyond the human annotation (necessarily at a cost of scale decrease compared to synthetic ones).

That being said, even these prior binary comparison frameworks themselves seem less central to how personalization is conceptualized today. Evaluating how models are able to generate personalized content as interaction continues with more context over the user seems like a more relevant topic. Given this shift, this work feels both insufficiently novel relative to prior binary-comparison work and misaligned with current research priorities.

3. The evaluation setup implicitly assumes that human annotators, when shown a persona and two candidate responses, can reliably “imagine themselves as the persona” and choose the better-aligned answer. However, this assumption is nontrivial: annotators may rely on superficial heuristics (e.g., keyword or style matching) that do not reflect the underlying user’s actual preference model. It is therefore unclear whether a higher score on this benchmark corresponds to being meaningfully better at real personalization, where user preferences are often inconsistent, evolving, or only partially observable.

**Questions:**

See above weakness

---

> ### Author Response · Authors · 2025-11-17
>
> ### Weaknesses:
>
> > Q1: The main concern is over the design of the benchmark, i.e., binary choice evaluation. In essence, the benchmark is measuring the model's ability to recognize personalization (which one of the two responses better reflects a persona), instead of measuring how good the model is at independently generating a high-quality personalized response. Binary discrimination is cognitively and computationally much easier than fluent personalization in open-ended dialogue. This seems a critical distinction in the current stage of LLM personalization research: personalization has moved beyond Alpaca-style preference tuning toward direct generation on user attributes, and the paper never addresses this.
>
> > While such comparison-based supervision played a historical role in preference alignment (e.g., Alpaca, DPO), it is hard to view it as a final personalization objective at this stage of time.
>
> Thank you for your feedback on the benchmark's design. We will clarify in the manuscript that evaluating the *selection* of a personalized response (a discriminative task) and the *generation* of one are distinct capabilities. Accordingly, we position PersonaFeedback not as a benchmark for open-ended generation, but as a tool to ​**measure a model's ability to judge response quality based on explicit personas**​. It provides a foundational component for assessing persona understanding, rather than addressing the full scope of long-term or implicit personalization.
>
> While a discriminative task may seem simpler,**​ our key finding reveals it is far from a solved problem.** State-of-the-art LLMs struggle significantly on our benchmark's 'hard' difficulty setting. This shows that when personalization is nuanced, this "simple" binary choice becomes a rigorous test of a model's deep comprehension, exposing a fundamental capability gap.
>
> **This discriminative approach also offers crucial diagnostic value.** In open-ended generation, it is difficult to disentangle a model's failure to understand a persona from its general generation flaws. Our work isolates the persona understanding aspect, providing a precise tool to diagnose this specific weakness, which is often obscured in generative evaluations.
>
> Finally, **a robust discriminator has immense practical value.** In real-world applications, generative models evolve rapidly, making continuous reliance on human evaluation unsustainable due to its high cost and slow pace. A powerful discriminator serves as a ​**stable and automated proxy for human judgment**​, enabling consistent quality assessment as models iterate.
>
> However, the reliability of such an automated judge is paramount. To validate its effectiveness and ensure its judgments align with human preferences, a standardized benchmark is essential. PersonaFeedback is designed precisely for this purpose. It provides a human-annotated, multi-level dataset to rigorously test and calibrate these discriminative models, ensuring their evaluations are both consistent and trustworthy.

---

> > ### Author Response · Authors · 2025-11-17
> >
> > > Q2: For these binary preference recognition type of benchmark / work, there has already been couple in the literature (i.e., Alpaca, Zollo et al 2024), and it is unclear to the reviewer how this benchmark is materially different from the existing ones beyond the human annotation (necessarily at a cost of scale decrease compared to synthetic ones).
> >
> > > That being said, even these prior binary comparison frameworks themselves seem less central to how personalization is conceptualized today. Evaluating how models are able to generate personalized content as interaction continues with more context over the user seems like a more relevant topic. Given this shift, this work feels both insufficiently novel relative to prior binary-comparison work and misaligned with current research priorities.
> >
> > We fully agree that evaluating a model's generative capabilities during continuous interaction is an important objective in the field. However, we argue that this complex, end-to-end interaction **conflates** two challenges that are functionally distinct and can be independently evaluated: (1) inferring a user profile from history, and (2) utilizing a *given* (known) profile to generate personalized content. Existing benchmarks mentioned by the reviewer (e.g., Zollo et al. 2024, LaMP, PersonaMem) primarily focus on the former challenge, as they require the model to *implicitly* infer user preferences from interaction history. The core contribution and novelty of PERSONAFEEDBACK lie in its design as the first benchmark to **decouple** these two challenges by providing an ​*explicit user profile*​. This design enables us, for the first time, to independently and precisely evaluate a model's core capability in *utilizing* a profile (i.e., Challenge 2) *without* this evaluation being *confounded* by the model's proficiency (or lack thereof) in inference (Challenge 1). Thus, our fundamental distinction from Alpaca or Zollo et al. is not merely the use of human annotation. First, we move beyond testing for alignment with **general human values** to measuring a model's ability to align with ​**specific, individual personas**​. Second, there is a **fundamental shift in evaluation methodology** (explicit profiles vs. implicit history) and the ​**introduction of difficulty stratification**​—both of which were absent in prior benchmarks. Furthermore, we adopted the binary comparison framework not as an end goal, but because it is currently the most reliable **diagnostic tool** for subjective tasks such as personalization. As discussed in our paper, frameworks that directly evaluate generated content (e.g., LLM-as-a-judge) face significant difficulties in distinguishing subtle, fine-grained differences and maintaining interpretability. Our human-annotated binary-choice task allows for a more effective quantification of subtle differences in model performance.
> >
> > > Q3: The evaluation setup implicitly assumes that human annotators, when shown a persona and two candidate responses, can reliably “imagine themselves as the persona” and choose the better-aligned answer. However, this assumption is nontrivial: annotators may rely on superficial heuristics (e.g., keyword or style matching) that do not reflect the underlying user’s actual preference model. It is therefore unclear whether a higher score on this benchmark corresponds to being meaningfully better at real personalization, where user preferences are often inconsistent, evolving, or only partially observable.
> >
> > Our annotation protocol (Appendix D) is designed precisely to mitigate this risk of superficial heuristics.
> >
> > Specifically, our protocol instructs annotators to **penalize** forced or unnatural answers—prioritizing *Helpfulness* alongside personalization—rather than rewarding superficial matching.
> >
> > Furthermore, our benchmark (Appendix H, Table 10) features "Over-personalization" examples where the *rejected* response is exactly the type of forced keyword-matching the reviewer mentioned. This confirms our benchmark directly tests a model's ability to ​**distinguish meaningful integration from mere surface-level matching**​.
> >
> > ### Acknowledgement
> >
> > We thank you for these detailed and constructive comments. They have helped us clarify the scope of PersonaFeedback, sharpen our comparison to prior binary-comparison benchmarks. We believe the revisions prompted by your feedback will make the paper’s contributions and boundaries more transparent to readers.

---

### Meta-Review · Area_Chair_mePh · 2026-01-11

**Summary:**

The paper introduces **PersonaFeedback**, a **human-annotated** benchmark with **8,298** pairwise test cases for evaluating *persona-conditioned personalization* in LLMs. The key design choice is to **decouple persona inference from personalization** by providing an **explicit persona profile** and asking models to **select** (binary choice) which of two candidate responses better matches the persona while remaining helpful. The dataset is stratified into **easy/medium/hard** tiers based on annotator agreement, and the paper reports broad evaluations across many model families (reasoning/chat/open-source/reward models) and different settings (explicit persona profile vs. RAG vs. no persona). The reported findings include: (i) stronger reasoning does not reliably translate to stronger personalization, (ii) scale correlates with better performance, (iii) naive RAG does not help and can underperform “no persona,” and (iv) even strong models still struggle on the hard tier; humans perform substantially higher.

### Strengths
- **Clear problem framing and decoupling goal:** Explicitly isolates the “use a given persona” component rather than conflating it with persona inference from interaction history.
- **Substantial human annotation and quality controls:** 8,298 human-labeled pairwise preferences; multiple annotators; agreement-driven difficulty stratification; filtering based on consensus thresholds; reported human expert baseline.

### Weaknesses
- **Scope mismatch / over-claim risk:** The benchmark is fundamentally **discriminative** (selecting the better response) rather than **generative** (producing a personalized response), so claims must be carefully scoped to “judging/recognizing persona-consistent responses,” not end-to-end personalization generation.
- **Binary-choice abstraction limitations:** Pairwise forced choice and accuracy as the sole metric simplify personalization, which is often graded and multi-dimensional; this narrows what “success” means.

While the rebuttal clarifies scope (benchmarking persona-conditioned discrimination/judging) and provides additional details (annotation protocol, RAG details, stronger RAG experiment, significance testing, human baseline, ablations), the fundamental concern remains: the benchmark does not directly measure a model’s ability to *generate* personalized responses, and the paper’s central contributions hinge on positioning this discriminative task as a primary evaluation target.

**Reviewer Concerns:**

**Concerns largely addressed by rebuttal**
- **RAG implementation details and “naive vs stronger RAG” clarification** (kmbr, krUJ): authors specify top-k=20, memory snippet lengths, FAISS+dense embeddings, and report a stronger RAG variant (LLM filtering/reconstruction) that still underperforms explicit persona profiles.
- **Need for statistical testing** (krUJ): authors add McNemar tests and a triple-sampling analysis on hard cases.
- **Ablations / more analysis requested** (krUJ): authors add persona-field ablations (showing preferences matter most) and question-category analysis; provide human expert baseline.
- **Annotator instructions / selection** (SEns): authors point to Appendix D with selection protocol and clarify criteria (helpfulness + personalization); plan to add a Limitations section.
- **Clarification of reward models / trained models** (SEns): authors explain which reward models were evaluated vs. small models trained using BT loss.

**Concerns still outstanding:**
- **Core task validity relative to claimed goal (discriminative vs. generative personalization)** (ZcZM, kmbr, SEns): rebuttal clarifies intended scope, but does not eliminate the central mismatch criticism—i.e., the benchmark evaluates *recognition/judgment* of persona-consistency rather than *production* of personalized responses, and reviewers question whether this is the most central/appropriate evaluation target for “LLM personalization.”
- **Novelty/positioning relative to prior pairwise preference frameworks** (ZcZM): authors argue novelty via explicit-profile decoupling and difficulty stratification; however, the reviewer’s core concern about the relevance/priority of binary preference recognition for personalization remains unresolved.

**Reviewer Scores:**

I expect that two strong rejects will remain a rating of reject while positive reviews won't champion for this submission.

---

### Decision · Program_Chairs · 2026-01-26

Reject